# New Downscaling Approach Using ESA CCI SM Products for Obtaining High Resolution Surface Soil Moisture

**Jovan Kovačević \***[ID]**, Željko Cvijetinović**[ID]**, Nikola Stančić, Nenad Brodić and Dragan Mihajlović**

Faculty of Civil Engineering, University of Belgrade, Bulevar kralja Aleksandra 73, 11000 Belgrade, Serbia; zeljkoc@grf.bg.ac.rs (Ž.C.); nstancic@grf.bg.ac.rs (N.S.); nbrodic@grf.bg.ac.rs (N.B.); draganm@grf.bg.ac.rs (D.M.)
\* Correspondence: jkovacevic@grf.bg.ac.rs

**Abstract:** ESA CCI SM products have provided remotely-sensed surface soil moisture (SSM) content with the best spatial and temporal coverage thus far, although its output spatial resolution of 25 km is too coarse for many regional and local applications. The downscaling methodology presented in this paper improves ESA CCI SM spatial resolution to 1 km using two-step approach. The first step is used as a data engineering tool and its output is used as an input for the Random forest model in the second step. In addition to improvements in terms of spatial resolution, the approach also considers the problem of data gaps. The filling of these gaps is the initial step of the procedure, which in the end produces a continuous product in both temporal and spatial domains. The methodology uses combined active and passive ESA CCI SM products in addition to in situ soil moisture observations and the set of auxiliary downscaling predictors. The research tested several variants of Random forest models to determine the best combination of ESA CCI SM products. The conclusion is that synergic use of all ESA CCI SM products together with the auxiliary datasets in the downscaling procedure provides better results than using just one type of ESA CCI SM product alone. The methodology was applied for obtaining SSM maps for the area of California, USA during 2016. The accuracy of tested models was validated using five-fold cross-validation against in situ data and the best variation of model achieved RMSE, $R^2$ and MAE of 0.0518 m$^3$/m$^3$, 0.7312 and 0.0374 m$^3$/m$^3$, respectively. The methodology proved to be useful for generating high-resolution SSM products, although additional improvements are necessary.

**Keywords:** soil moisture; downscaling; random forest; ESA CCI SM

## 1. Introduction

Soil moisture is a crucial component in Earths' system with great impact on interactions between the land surface and the atmosphere [1]. Consequently, using soil moisture information is critical to many applications such as hydrogeological monitoring [2,3], meteorology [4] and water resource management [5,6]. Soil moisture also plays an important role in evapotranspiration process [7], which subsequently influences precipitation occurrences [8]. Soil moisture also indirectly affects environment where its relationship with forest fires has been recognized [9]. Importance of soil moisture has also been recognized institutionally as it is listed as one of the 50 Essential Climate Variables within the Global Climate Observing System (GCOS) [10,11].

Soil moisture can be defined as a mass or volume of water stored between the earth particles in the upper unsaturated soil layer. It is usually distinguished as the surface soil moisture (SSM), which represents the topsoil water content (0–5 cm depth), and the root zone soil moisture (RSM), which accounts for water available to the plants' root system (<2 m depth) [12,13]. Soil moisture

content is traditionally measured using ground instruments and techniques based on: (1) sampling and drying; (2) electrical resistance; (3) neutron scattering; (4) gamma-ray absorption; or (5) time-domain reflectometry [12]. This way, both SSM and RSM can be obtained in a form of point measurements and their spatiotemporal characteristics over the wider area have to be modeled, usually using geostatistical methods [14–16]. With the advancements of the satellite remote sensing, an alternative method for the retrieval of the soil moisture came to attention. Satellite observations provided a way of obtaining soil moisture content over the regional and global scales with the temporal resolution in a matter of days. Based on the part of the electromagnetic spectrum being used, the following satellite sensors proved to be useful for soil moisture mapping: (1) microwave (active and passive); (2) optical; and (3) thermal [1]. Unfortunately, due to the penetration depth of the electromagnetic waves through the soil, only SSM can be obtained from the satellite remote sensing [17], while RSM has to be obtained through vertical extrapolation [18]. Very comprehensive and recent reviews on possibilities of generating SSM from the satellite remote sensing data were done by Sabaghy et al. [19] and Peng et al. [20].

Numerous microwave remote sensing sensors have been developed and used for mapping soil moisture content. These include the Advanced Microwave Scanning Radiometer—Earth Observing System (AMSR-E) [21], Soil Moisture and Ocean Salinity (SMOS) satellite [22], Soil Moisture Active Passive (SMAP) mission [23], the Advanced Scatterometer (ASCAT) [24], ESA Sentinel-1 satellites [25] and many more. To achieve the optimal temporal and spatial coverage and to produce the long time series of soil moisture data, all these sources need to be synchronized and merged in the data assimilation process. During this procedure, the differences in operational, spatial, temporal and retrieval algorithm aspects of the used sources must be taken into account. The European Space Agency (ESA) produces such merged microwave soil moisture products as part of the Climate Change Initiative (CCI)—ESA CCI SM [26]. Although the ESA CCI SM product provides very good spatial coverage, there are still data gaps in some places. Another disadvantage is that the product has coarse spatial resolution of 0.25° (≈25 km), which is insufficient for many regional and local applications.

Several studies have aimed at improving the spatial resolution and filling the data gaps of coarse resolution SSM products [27–29]. Machine learning (ML) techniques proved to be a very useful tool for such purpose [30,31]. Studies have shown that Random Forest (RF) is one of the many available ML techniques that yields very good results in downscaling and filling data gaps thanks to its flexibility through randomization and ensemble approach [32]. This study successfully implemented a two-step approach to produce SSM product without missing data and with high spatial resolution (1 km). The first step is used as a data engineering tool and its output is used as the input for the second step. Bilinear interpolation and random forest model are considered as data engineering tools in the first step, and, in the second step, additional random forest regression is used. The methodology was tested over the study area of California, USA for the year 2016. ESA CCI SM products, together with the auxiliary products (Normalized Difference Vegetation Index (NDVI), Land Surface Temperature (LST), NWS Precipitation, and Köppen–Geiger climate classification map), were used within the prediction model. The approach described in this paper is novel in a method that considers the synergic use of multiple ESA CCI SM products instead of a single one in order to obtain high resolution SSM maps.

## 2. Materials

### 2.1. Study Area

The study area covers 423,967 $km^2$—the complete state of California, USA (Figure 1). The area's relief is dominated by the Central Valley, which runs 725 km through the state between the Coast Ranges to the west and the Sierra Nevada to the east and bounded by the Cascades in the north and Tehachapi Mountains to the south. California's land cover is diverse, where forests cover almost half of the state's area and with barren plains in the northern and desert area in the east-central parts. Climate conditions in California vary from polar to subtropical. The biggest part of the state has a Mediterranean climate and in the northeastern part the temperate climate is present. The climate

also changes rapidly with elevation, where the alpine climate can be found in the higher mountains. Different parts of the state receive various amount of precipitation, which ranges from more than 4300 mm in the northwest to small traces in the southeastern desert. Coastal areas are different too, where moderate temperatures and moderate rainfall prevail.

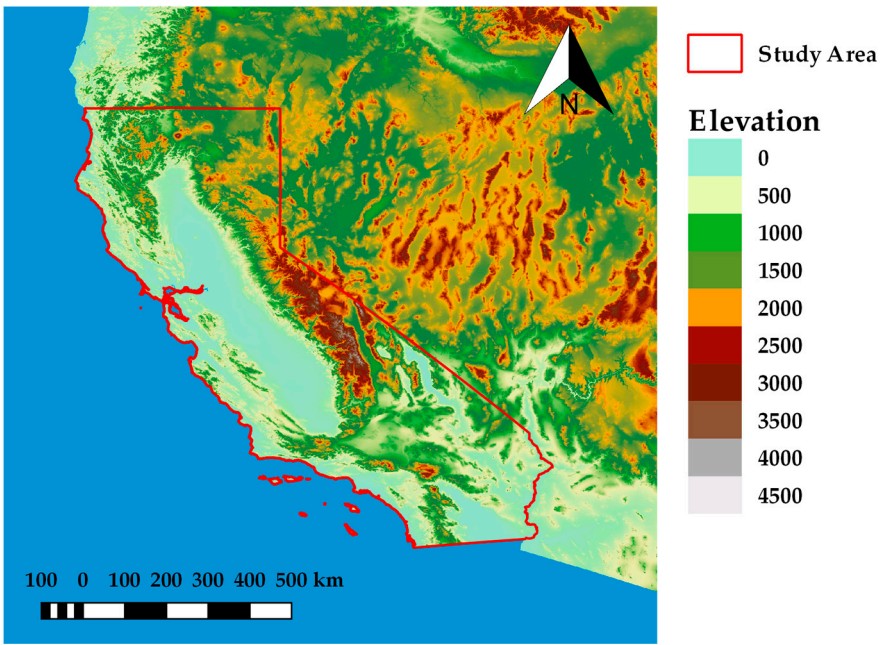

**Figure 1.** The study area—the state of California, USA.

The state is a major agriculture contributor accounting for over 13% of the USA's total agricultural value in 2018. It produces more than 400 commodities, with more than a third of the country's vegetables and two-thirds of the country's fruits and nuts being grown in California [33]. Such extensive agriculture production requires careful and smart water management, which can benefit significantly from high quality soil moisture maps.

## 2.2. European Space Agency Soil Moisture Products - ESA CCI SM

ESA CCI SM products are generated using soil moisture observations from active (ERS1-2 SCAT and MetOp ASCAT A-B) and passive (SMMR, SSM/I, TMI, AMSR-E, WindSat, AMSR2 and SMOS) microwave satellite sensors. Three groups of soil moisture products are generated in the assimilation process: active (ESA CCI SM A), passive (ESA CCI SM P) and combined (ESA CCI SM C). The active soil moisture products are generated from the C-band scatterometers using the change detection algorithm. The passive products are handled using the Land Parameter Retrieval Model (LPRM), which successfully translates the microwave observed land surface brightness temperature (Tb) to the soil moisture content. The combined product is obtained through the assimilation process of the previous two, with the appropriate weights assigned to each source [26]. All products provide daily global coverage with the spatial resolution of 0.25° (≈25 km). Active soil moisture products are expressed in the percentage of saturation (%), whereas passive and combined soil moisture products are expressed in volumetric units ($m^3/m^3$). In the latest version of ESA CCI SM products (04.5), the temporal range has been extended and covers 1978–2018. In this research, all three types of products (passive, active and combined) for 2016 were obtained from the ESA data archive (https://www.esa-soilmoisture-cci.org/).

## 2.3. PBO_H2O in Situ Soil Moisture Observations

PBO_H2O, a project that was operational from 2004 to 2017, implemented GPS interferometric reflectometry for the measurement of SSM. The observations represent volumetric soil moisture

content in the topsoil layer (0–5 cm) with spatial scale of ~1 km$^2$ and accuracy of 0.04 m$^3$/m$^3$ [34]. PBO_H2O data can be obtained from the International Soil Moisture Network (ISMN) data archive (https://ismn.geo.tuwien.ac.at/en/), as it was done for the whole 2016. The complete dataset consists of 159 stations with hourly measurements. For each station, the observations were firstly aggregated to obtain the mean daily value. In the next step, the locations with the multiple sensors (same latitude and longitude) were averaged. fifty-six were stations left after cropping locations to the study area (Figure 2a), with a total of 18,307 daily surface soil moisture observations (Figure 2b).

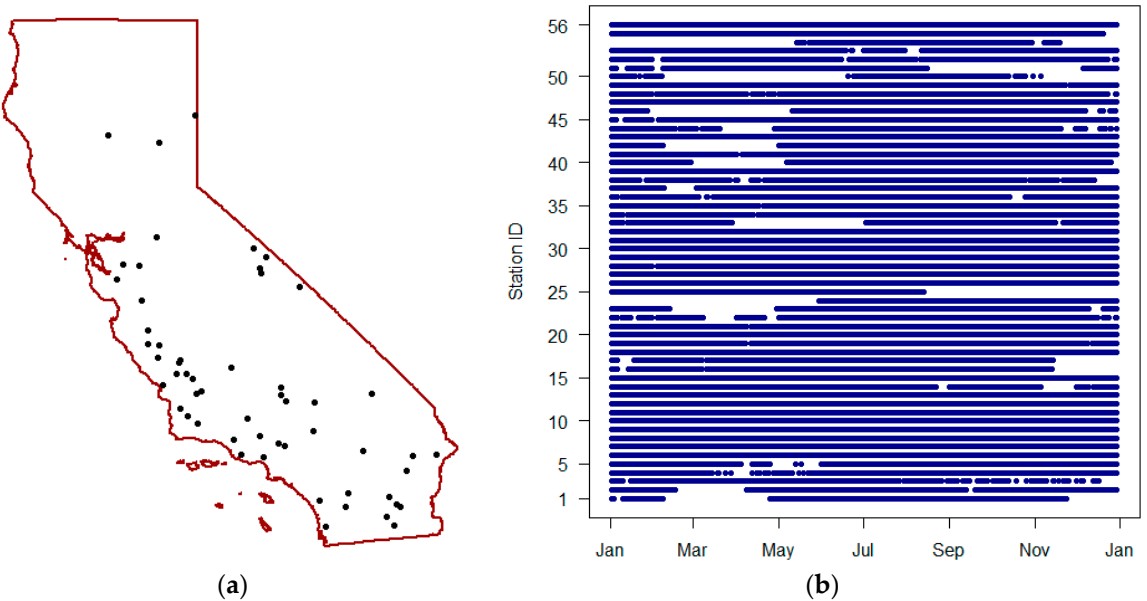

**Figure 2.** (**a**) Spatial distribution of the PBO_H2O soil moisture stations in the study area; and (**b**) soil moisture observations per station during the 2016.

## 2.4. Auxiliary Data

### 2.4.1. Normalized Difference Vegetation Index (NDVI) and Land Surface Temperature (LST)

The connection between land water content and NDVI and LST has been widely used for downscaling coarse resolution remotely-sensed soil moisture [35–38]. The main advantage of using such data for downscaling is their fine spatial resolution, good temporal coverage and the many available satellite missions that collect them. However, the cloud contamination is a big problem for all optical sensors, making these products unavailable in certain places [1]. Moderate Resolution Imaging Spectroradiometer (MODIS) is one of the most commonly used sources for such products and therefore it was chosen as the provider of NDVI and LST.

NDVI was taken from MODIS Vegetation Indices 16-day Level 3 Global 1 km Version 6 products, from both Terra (MOD13A2) and Aqua (MYD13A2) satellites. The temporal coverage of NDVI included 2016 and 2017 with 46 Terra and 46 Aqua products. Each product was generated in WGS84 coordinate reference system. The data coverage was extended to include 2017 because it was necessary for generating and later improving daily NDVI products.

The LST data were induced from MODIS Land Surface Temperature/Emissivity Daily L3 Global 1km Version 6 products from both Terra (MOD11A1) and Aqua (MYD11A1) satellites. LST$_{DAY}$ and LST$_{NIGHT}$ land surface temperature maps in the form of rasters in WGS84 coordinate reference system were generated for each satellite and for each date of 2016 (ideally, four rasters for each date). In the next preprocessing step, for each date, Terra and Aqua products were merged by taking average of corresponding pixels, so that, in the end, single LST$_{DAY}$ and LST$_{NIGHT}$ rasters were produced for each

date of 2016. Since the data for Terra products DOY 50-58 were missing, only the Aqua products were used for producing LST$_{DAY}$ and LST$_{NIGHT}$ rasters during these days.

### 2.4.2. NWS Precipitation Data

As a part of the natural water cycle, atmospheric water is transferred to the land through precipitation. The correlation between precipitation and soil moisture spatial and temporal patterns has been observed by many studies [2,8]. Since precipitation datasets are of higher spatial resolution, it has been used in the process of downscaling coarse resolution soil moisture [32,39].

National Weather Service (NWS) produces daily precipitation estimate maps for the whole USA from the combined sensor inputs: radar and rain gauge. The data represent 24-h accumulation and they are disseminated in the Hydrologic Rainfall Analysis Project (HRAP) grid coordinate system. Although the spatial resolution of the data is considered roughly ≈4 km over continental USA, the spatial resolution of the product over the study area is closer to ≈5 km due to the characteristics of the HRAP grid. After obtaining the data for 2016 (https://water.weather.gov/precip/), each file was preprocessed to ≈0.05° (≈5 km) in WGS84 coordinate reference system. Each HRAP grid point was assigned to the closest WGS84 pixel during the preprocessing.

### 2.4.3. Köppen–Geiger Climate Classification Map

Climate types are defined using average weather conditions over a long time. The certain climate type is directly or indirectly related to the precipitation amount, the dominant vegetation density/types and the land surface temperature [2]. Therefore, it can be expected that it can be useful for the downscaling procedure. To the authors' knowledge, no other studies used climate data for downscaling soil moisture.

Köppen–Geiger climate classification map is the most frequently used climate classification map created by Wladimir Köppen and it was presented in its latest version in 1961 by Rudolf Geiger. In this research, the updated and re-analyzed Köppen–Geiger map produced by Climate Change & Infectious Diseases was used [40]. The spatial resolution of the map is 5' and it can be obtained from the group's website (http://koeppen-geiger.vu-wien.ac.at/). The climate classification map was additionally reclassified to first level of classification scheme with five different climate groups: A (Tropical), B (Arid), C (Temperate), D (Continental) and E (Polar).

## 3. Methods

### 3.1. Bilinear Interpolation

Bilinear interpolation is a widely popular two-dimensional interpolation method that uses the values of four closest points in order to estimate an output value [41]. The interpolation function that is used to fit a bilinear surface through these four points is given by the equation:

$$z = f(x, y) = a_0 + a_1 x + a_2 y + a_3 xy. \tag{1}$$

When applied to a raster image, this interpolation method considers the known values of the four nearest pixels located in diagonal directions from the position of a new pixel. A new pixel value is calculated as a weighted average of these four pixel values from the original image. This resampling method can be used both as an aggregation or disaggregation raster tool. In this research, it was considered as a disaggregation tool used for downscaling remote sensing products from coarse to finer spatial resolution. Due to its vast popularity, the bilinear interpolation was taken for comparison purposes, that is, to compare its results with the results of the methods that are more sophisticated.

### 3.2. Random Forest Regression

Random forest is an ensemble approach machine learning technique which can be applied for both regression and classification problems. The technique proposed by Breiman [42] uses multiple decision trees built during the training phase from which mean prediction is taken as an output of the model. Each tree is built from the bootstrap sample created from some portion of the input training data, while the remaining data are used for the performance evaluation of each tree. This feature (also known as bootstrap aggregation) provides powerful tool for modeling nonlinear relationships while reducing the chance of overfitting and improving generalization [42].

In this study, random forest regression implemented in ranger R package was used [43]. The number of trees was set to 200 because a larger number did not produce significant error improvement, but increased the computation time. The split rule was set to "MaxStat" instead of the more usual default "Variance" split rule. All other parameters were left to their default values.

## 4. Methodology

The methodology used in this research consists of several steps (Figure 3). First, the input datasets were processed to fill gaps in the data in both temporal and spatial domains. Next, the created datasets were used to downscale coarse resolution ESA CCI SM products to high spatial resolution of 1 km (Data engineering). Since downscaled products still have large bias against the in situ soil moisture observations, additional processing was necessary. This was covered in the final step (Random forest), where all previously created downscaled datasets in congregation with in situ data were used to produce output SSM maps of high spatial resolution. The following sections describe all these steps in detail.

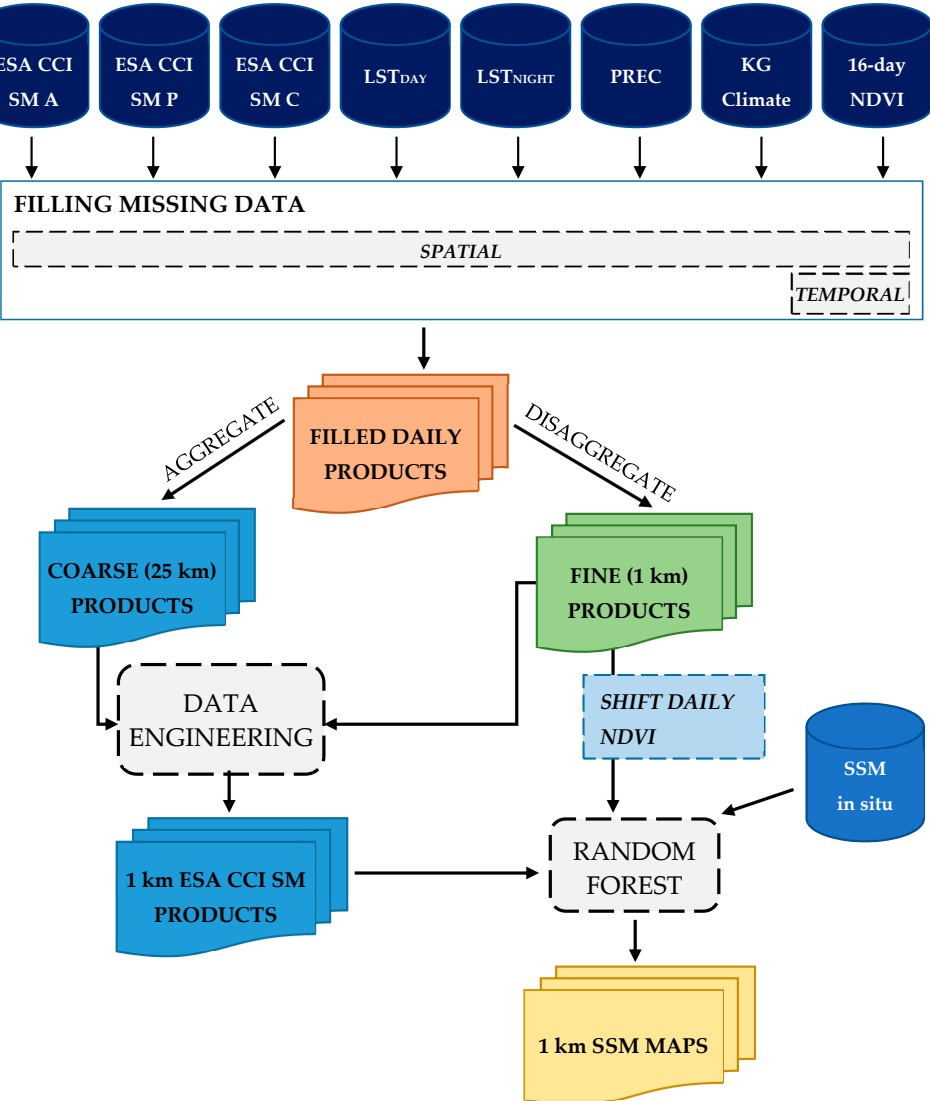

**Figure 3.** The flowchart diagram of the data processing steps.

*4.1. Filling Spatial and Temporal Data Gaps*

All input raster datasets (except climate classification map) have some spatial gaps. Gaps in ESA CCI SM products are caused by the lack of microwave soil moisture sources and their spatial coverage for some specific day; gaps in MODIS datasets are caused by clouds and/or other atmospheric conditions; and NSW precipitation has some missing data left after the transformation from HRAP grid to WGS84. To fill all missing data pixels in the study area, universal kriging interpolation technique is used. Universal kriging showed good performances compared to other commonly used interpolation techniques, almost as good as kriging with an external drift [44]. The advantage is that universal kriging does not require additional variables within the interpolation process. This enables that each of the input datasets can be filled independently of the other datasets.

The sample variogram is generated and used for fitting the spherical variogram model, where each raster pixel is considered as observation point. For the computational effectiveness, sample variogram was modeled using the 0.25° (≈25 km) spatial resolution, meaning that all datasets that have different spatial resolution (LST, NDVI and NWS Precipitation) have to be aggregated by the mean value before the variogram modeling. Using this technique, missing data for each input raster are independently filled. Additionally, for each NDVI 16-day composite raster, the raster that represents the day of

the year that NDVI pixel corresponds to is generated and its spatial gaps are filled (NDVI_DOY). NDVI_DOY pixel values are rounded to avoid decimal values.

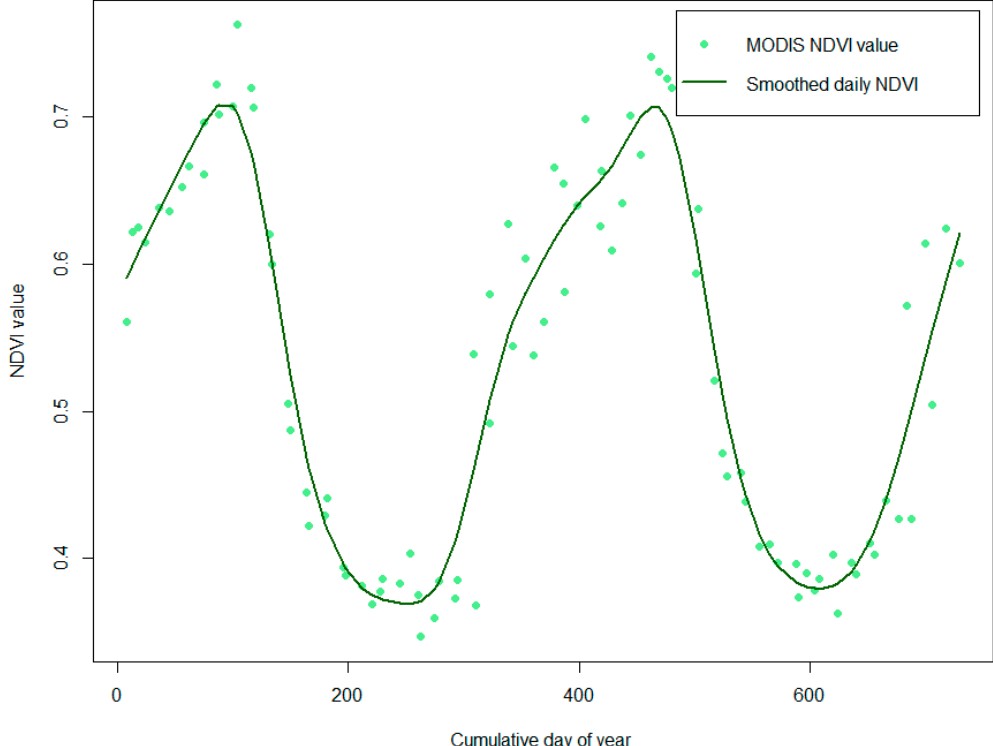

**Figure 4.** An example of the fitted cubic smoothing spline for NDVI temporal gap filling for the period 2016–2017.

Temporal gaps are a big problem for NDVI data, which are 16-day composites. Even with both satellites used synergically, the NDVI observations are ideally available every 9 days, which is too sparse for modeling daily SSM. The smoothing methods provide relatively simple, yet effective way for reconstructing NDVI time-series [45]. No smoothing method can be recommended more than others. However, spline smoothing provides rather good results and its parameters can be well tuned through cross-validation [45]. Therefore, temporal gap filling is done using NDVI and NDVI_DOY information pixel-wise, by fitting a cubic smoothing spline. Since no ground NDVI dataset is available for determining the optimal spline parameters through cross-validation, these parameters were determined empirically by visual inspection of the smoothing curves. The curve is fitted in a way that the changes of NDVI are gradual, without unusual spikes or drops (Figure 4). This approach leads to smooth NDVI. Although this can lead to a smooth soil moisture time series, it is expected that such behavior will be avoided by the use of other daily available predictors. The values for each day are then generated after the spline fitting.

Correlation between Daily Filled Predictors and in Situ Soil Moisture Observations

Before proceeding to the next step, the relevance and quality of each daily filled predictor is assessed. This is done by calculating Pearson correlation coefficient between available in situ soil moisture observations and the daily filled values of the predictors to be used in the prediction model. As shown in Figure 5, strong positive correlation exists between the in situ soil moisture and all three types of ESA CCI SM products along with NDVI. Among others predictors, only $LST_{DAY}$ shows strong negative correlation, while all others ($LST_{NIGHT}$, PREC and Climate) show medium correlation with the in situ data. Such correlation values indicate that filling data gaps was successful. Since at least the medium correlation exists, using the chosen set of predictors in the downscaling procedure is justified.

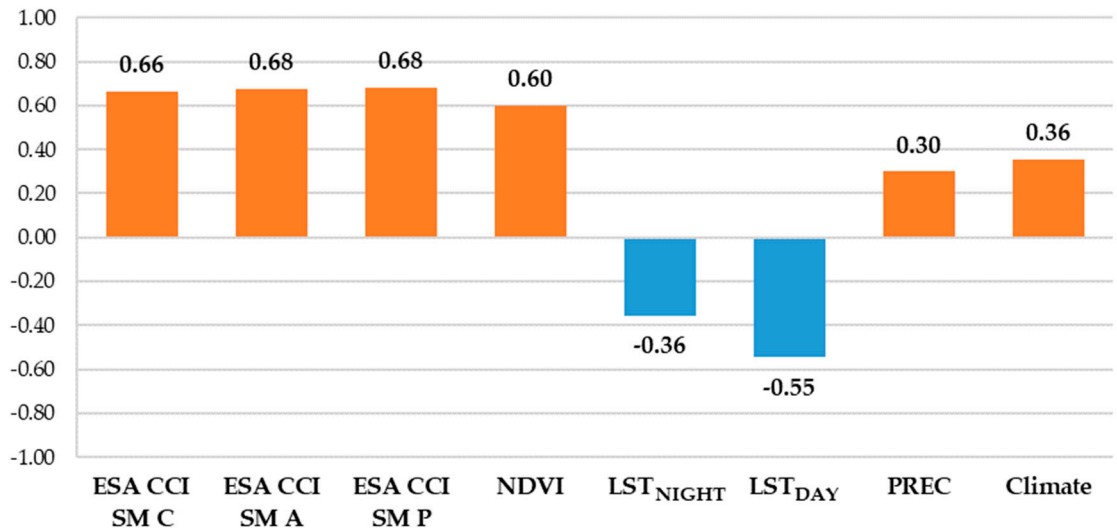

**Figure 5.** Pearson correlation coefficients between in situ soil moisture observations and each predictor.

### 4.2. Downscaling ESA CCI SM Products

Downscaling of the ESA CCI SM products is done using the previously generated datasets. No reprojection is necessary since all data are already in the common coordinate reference system—WGS84. The downscaling is performed independently using the bilinear interpolation technique (BIL) (Data engineering) and using the random forest (RF) method. The RF model (Data engineering) is defined as:

$$ESA_{SM-D} = DOY + ESA_{SM-1} + ESA_{SM-2} + NDVI + LST_{DAY} + LST_{NIGHT} + PREC + Climate \quad (2)$$

where $ESA_{SM-D}$ is the downscaled ESA CCI SM product, $ESA_{SM-1}$ and $ESA_{SM-2}$ are the remaining two types of ESA CCI SM products and DOY, PREC and Climate represent day of the year, amount of precipitation and climate zone, respectively. The RF regression model is trained over the coarse spatial resolution of 25 km where all NDVI, $LST_{DAY}$, $LST_{NIGHT}$ and PREC rasters are aggregated firstly. The trained model is then applied for the generation of 1 km ESA CCI SM rasters using two other ESA CCI SM products, NDVI, $LST_{DAY}$, $LST_{NIGHT}$, PREC and Climate 1 km predictors. In cases where there are no 1 km predictors available (ESA CCI SM products, PREC and Climate), they are disaggregated from coarser to the desired 1 km spatial resolution. For ESA CCI SM products and PREC raster, this is done using standard bilinear interpolation. Considering that the climate raster is the categorical raster map, it is disaggregated to 1 km spatial resolution using the nearest neighbor interpolation.

### 4.3. Generating Surface Soil Moisture Maps of High Spatial Resolution

#### 4.3.1. Shifting NDVI Values

Figure 4 shows that there are some differences between observed and modeled NDVI values due to the spline fitting. The SSM has shifted, i.e., a delayed effect on vegetation, with the time lag of about half a month [35]. It can be expected that spline smoothing further emphasizes the delayed effect of SSM and NDVI. Therefore, even though it is useful for downscaling ESA CCI SM products, such NDVI product might not match well with the in situ soil moisture observations. Although the strong correlation (0.60) already exists between the in situ soil moisture observations and the daily filled NDVI, the removal of the time shift caused by smooth spline should further increase it.

Using this approach, the best shift value has been determined for each available in situ location. It is assumed that the best shift is represented by the shift value that corresponds to the highest correlation value between NDVI and in situ soil moisture observations (Figure 6a) obtained for each station independently. A range of shifts between −45 and +5 was tested against in situ soil moisture

observations. The final shift value for the whole study area was then determined as a median value of all individual best shifts (Figure 6b). In some cases, the shift value did not converge to the local minimum (its value corresponded to the edge shift values). Therefore, these shift values were omitted from the median calculation. Because the shift is expected to be negative (soil moisture content affects the vegetation in the future), the NDVI data for both 2016 and 2017 have to be used. It should be noted that correlation of NDVI and in situ changes monthly, thus it always has to be calculated for the same time interval, in order to make correlation values comparable over the shifting range. That is why the calculation is always calculated for 2016, no matter the shifting range being examined.

The final shift value for the study area determined using the previously explained method is -24 days. This is larger than the reported time shift, probably due to the smoothing effect. This way, the correlation of the shifted NDVI was increased to 0.65, almost as high as of the ESA CCI SM products.

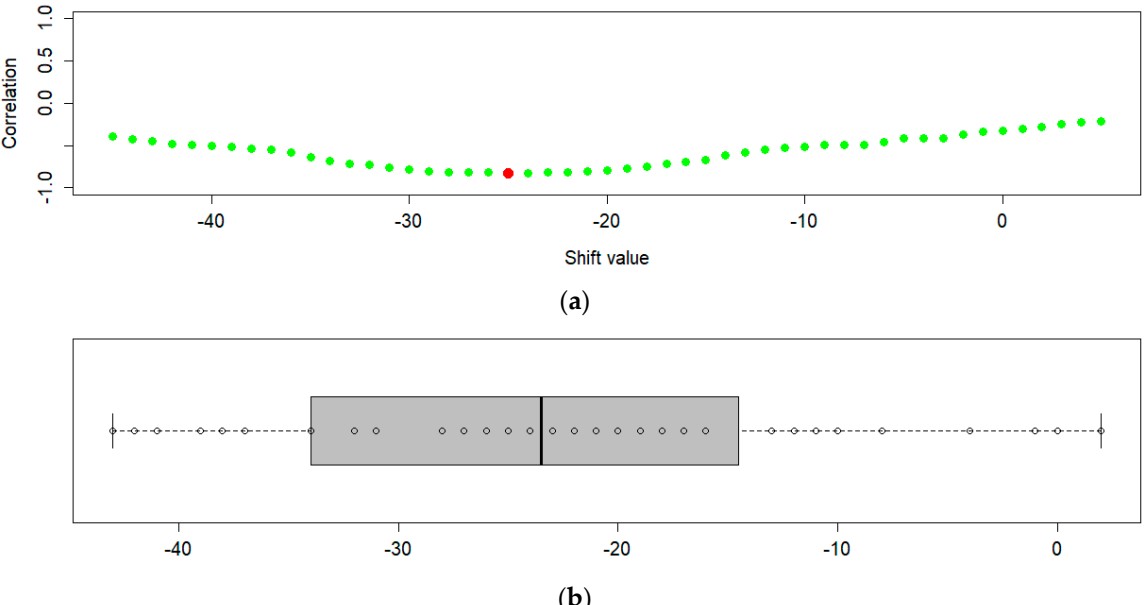

**Figure 6.** (**a**) An example of the determined best shift value compared to all correlation values from tested shift range; and (**b**) boxplot chart with individual best shift values from which the global shift value has been determined as median value.

### 4.3.2. Training Second RF Model

The second RF model uses all previously generated 1 km datasets, in addition to in situ soil moisture observations. The model (Random forest) is defined as:

$$SM_{in\ situ} = DOY + ESA_{SM-down} + NDVI_{SHIFT} + LST_{DAY} + LST_{NIGHT} + PREC + Climate \qquad (3)$$

All combinations of ESA CCI SM products were examined to determine the optimal one. Every model was trained using all available surface soil moisture observations and corresponding predictors for each location and each date. Because in situ soil moisture observations and the data for all predictors have the same scale, it is possible to implement extracting by using the value of the pixel that in situ location falls within. After successfully training the RF model, it was used to produce 1 km soil moisture maps for each day of 2016.

### 4.4. Validation of the Results

The model validation was performed using five-fold cross-validation, where complete fold locations were left out of the model training set and were only used for the model validation. Five-fold cross-validation was repeated 10 times and the output predictions were calculated as the mean values.

These were further used to determine the validation metrics. The metrics included root mean square error (RMSE), coefficient of determination ($R^2$) and mean absolute error (MAE) calculated between the observed soil moisture and the soil moisture generated from the cross-validation model output.

## 5. Results

The downscaling RF model (Data engineering) was trained using aggregated 25-km products for 2016. Since the data have no gaps, the 243,024 data entities (pixel stacks) are available for building the model. Two combinations were tested, one with and one without using other two ESA CCI SM products in the downscaling models. The version without using other two ESA CCI SM products was used as a benchmark, to determine the produced effect which these two products bring into the prediction model. All downscaled ESA CCI SM products were compared to the in situ data, where the passive product proved to be the best one by all metrics (displayed in Table 1).

**Table 1.** The validation metrics of downscaled ESA CCI SM 1-km products obtained by using in situ observations.

| | Downscaled Using Standard Bilinear Interpolation | | | Downscaled without Using Other Two ESA CCI SM Products | | | Downscaled Using Other Two ESA CCI SM Products | | |
|---|---|---|---|---|---|---|---|---|---|
| | RMSE [$m^3/m^3$] | $R^2$ | MAE [$m^3/m^3$] | RMSE [$m^3/m^3$] | $R^2$ | MAE [$m^3/m^3$] | RMSE [$m^3/m^3$] | $R^2$ | MAE [$m^3/m^3$] |
| ESA CCI SM C | 0.0745 | 0.4930 | 0.0591 | 0.0707 | 0.6095 | 0.0556 | 0.0694 | 0.6273 | 0.0557 |
| ESA CCI SM A [1] | / | 0.4860 | / | / | 0.5816 | / | / | 0.6199 | / |
| ESA CCI SM P | 0.0728 | 0.5092 | 0.0543 | 0.0613 | 0.6280 | 0.6280 | **0.0591** | **0.6495** | **0.0434** |

[1] RMSE and MAE cannot be determined because of the different unit system.

The second RF model was trained using all previously created daily 1 km predictors (ESA CCI SM products, NDVI, $LST_{DAY}$, $LST_{NIGHT}$, PREC and Climate) and 18,307 surface PBO_H2O soil moisture observations. All three versions of the downscaled ESA CCI SM products (without mixing them) were tested to determine the optimal combination. Since the downscaling step also introduces errors (see Table 1); testing all combinations helps understand the way these errors propagate in the following steps. The extracted validation metrics from the five-fold cross-validation are presented in Table 2.

**Table 2.** Validation metrics for each set of tested predictors (results after applying second RF).

| Predictor Combination [1] | | Without NDVI Shift | | | Using NDVI Shift | | |
|---|---|---|---|---|---|---|---|
| | | RMSE [$m^3/m^3$] | $R^2$ | MAE [$m^3/m^3$] | RMSE [$m^3/m^3$] | $R^2$ | MAE [$m^3/m^3$] |
| *Downscaled using bilinear interpolation* | *C+A+P* | 0.0528 | 0.7195 | 0.0382 | 0.0528 | 0.7191 | 0.0384 |
| | *C+A* | 0.0539 | 0.7101 | 0.0389 | 0.0538 | 0.7110 | 0.0390 |
| | *C+P* | 0.0543 | 0.7075 | 0.0391 | 0.0541 | 0.7089 | 0.0391 |
| | *A+P* | 0.0522 | 0.7294 | 0.0378 | 0.0521 | 0.7301 | 0.0378 |
| | *C* | 0.0560 | 0.6877 | 0.0403 | 0.0556 | 0.6916 | 0.0401 |
| | *A* | 0.0544 | 0.7058 | 0.0393 | 0.0543 | 0.7068 | 0.0394 |
| | *P* | 0.0546 | 0.7055 | 0.0392 | 0.0543 | 0.7087 | 0.0392 |
| *Downscaled using RF1 without other two ESA CCI SM products* | *C+A+P* | 0.0569 | 0.6729 | 0.0406 | 0.0561 | 0.6822 | 0.0402 |
| | *C+A* | 0.0568 | 0.6760 | 0.0404 | 0.0561 | 0.6843 | 0.0400 |
| | *C+P* | 0.0568 | 0.6763 | 0.0404 | 0.0561 | 0.6846 | 0.0400 |
| | *A+P* | 0.0566 | 0.6781 | 0.0405 | 0.0561 | 0.6837 | 0.0403 |
| | *C* | 0.0574 | 0.6705 | 0.0407 | 0.0566 | 0.6796 | 0.0403 |
| | *A* | 0.0574 | 0.6710 | 0.0410 | 0.0567 | 0.6779 | 0.0407 |
| | *P* | 0.0573 | 0.6717 | 0.0410 | 0.0567 | 0.6776 | 0.0408 |
| *Downscaled using RF1 with other two ESA CCI SM products* | *C+A+P* | 0.0525 | 0.7217 | 0.0379 | 0.0523 | 0.7238 | 0.0379 |
| | ***C+A*** | **0.0521** | **0.7279** | **0.0375** | **0.0518** | **0.7312** | **0.0374** |
| | *C+P* | 0.0521 | 0.7286 | 0.0375 | 0.0519 | 0.7305 | 0.0375 |
| | *A+P* | 0.0537 | 0.7104 | 0.0388 | 0.0536 | 0.7115 | 0.0387 |
| | *C* | 0.0524 | 0.7276 | 0.0376 | 0.0520 | 0.7310 | 0.0375 |
| | *A* | 0.0544 | 0.7051 | 0.0392 | 0.0541 | 0.7078 | 0.0390 |
| | *P* | 0.0543 | 0.7068 | 0.0391 | 0.0542 | 0.7068 | 0.0392 |

[1] C, A and P represent combined, active and passive ESA CCI SM products, respectively.

The improvements after each processing step can be clearly seen in Figure 7. The step with filling missing data produces continuous product, with both smaller and larger areas of missing data successfully reconstructed. As expected, the gap filling is less successful over the large missing data areas, where SSM content does not show reasonable changes. This is successfully covered in the next steps, where the gaps are hardly noticeable in the downscaled products, while they cannot be identified at all in the final products. Downscaling by BIL mostly fails to provide new spatial information. On the contrary, the main improvement of the downscaling process by RF is the spatial richness that has been obtained. Fine details which were previously hidden behind the coarse resolution can be differentiated in both the downscaled and the final output products. The local extremes present in the coarse resolution products are successfully adjusted in the following steps, albeit there are differences in soil moisture content between the downscaled and the final output products. Visually inspected, both types of downscaling, and especially bilinear interpolation, reduce extremes and produce smoother soil moisture products, while the in situ modeled products emphasize abrupt changes. This can be attributed to the bias that exists between the remotely-sensed soil moisture and in situ observations. As the ESA CCI SM products proved to be the most correlated predictor, the bias is successfully adjusted only when in situ observations are included in the model. This is done in the second RF regression model.

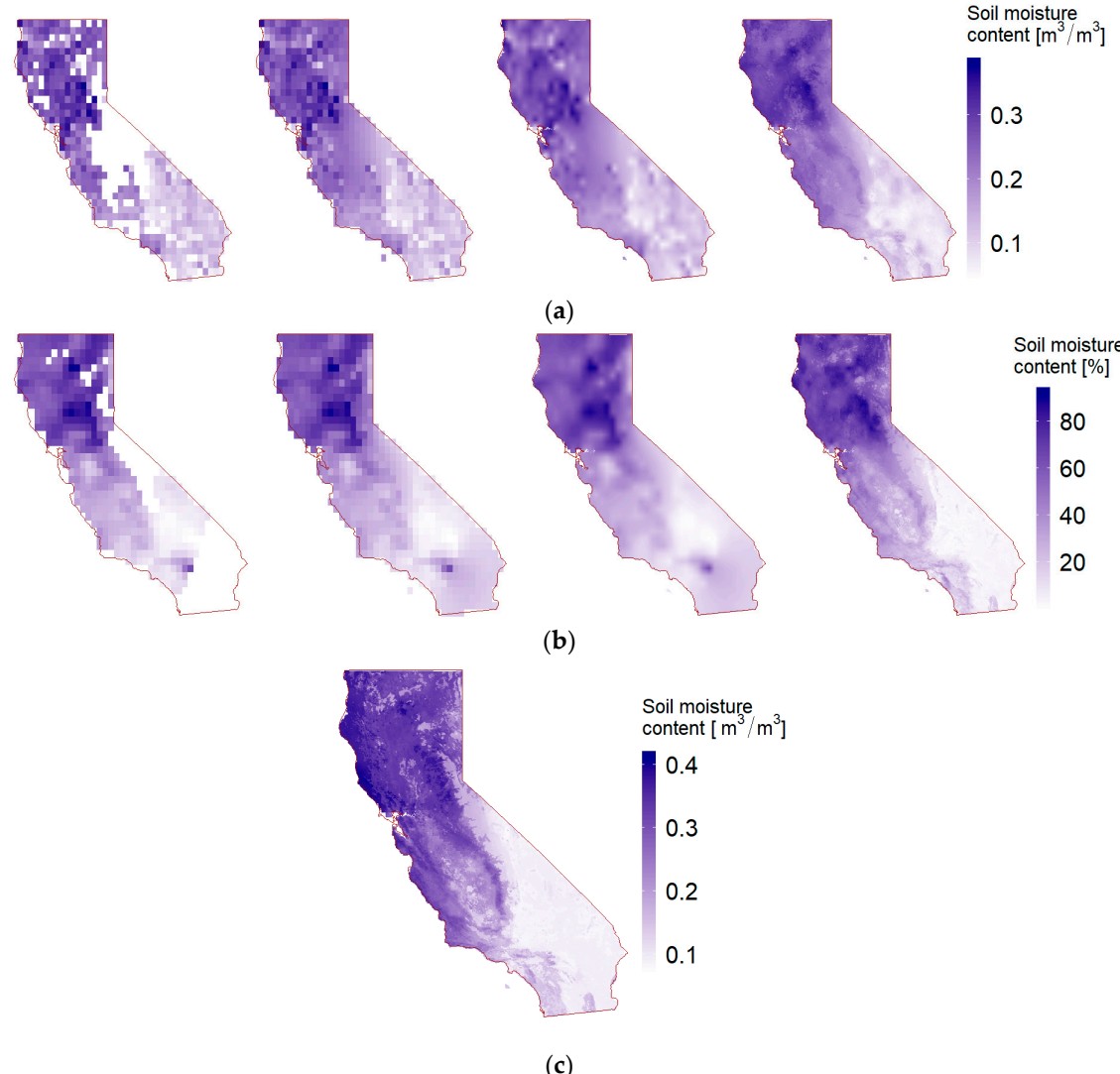

**Figure 7.** Output products after each processing step for DOY 81 2016: (**a**) combined ESA CCI SM input, daily filled, downscaled by bilinear interpolation (BIL) (Data engineering) and by RF products; (**b**) active ESA CCI SM input, daily filled, downscaled by BIL and by RF; and (**c**) final output SSM product.

## 6. Discussion

### 6.1. Determining the Best ESA CCI SM Predictor Combination

The supremacy of downscaling over standard bilinear interpolation as data engineering tool is as expected. Both types of RF models provided improvements across all validation metrics. However, there are still large errors present in all three types of spatial improvements. Incorporating ESA CCI SM products within the model leads to better performances across all metrics. Passive ESA CCI SM downscaled products outperform the others, but it still cannot be said with certainty which predictors should be chosen as an input for the second random forest model.

As displayed in Table 2, using RF downscaled products does not always contribute to a better prediction model. Although downscaling ESA CCI SM products using RF appeared to be a superior solution compared to the bilinear interpolation, it turns out that this is not always true. The use of first-step random forest only leads to marginally improved (or even deteriorated) accuracy compared to the bilinear interpolation method. The use of the RF downscaling procedure that excludes ESA CCI SM products provides the worst results across all metrics. In this case, there are no significant

differences regarding the set of used products in the second RF model. Using bilinear interpolation method outperforms all variants of RF downscaling procedure that excludes ESA CCI SM products. Some of the best results from all the tested combinations are provided this way. Out of all model variants that use bilinearly interpolated products, the variant that uses the combination of active and passive ESA CCI SM products provides the best accuracy. Such combination provides the third best of all metrics from variants without NDVI shift and the fourth best from variants that use NDVI shift. The use of the downscaling through RF outperforms the bilinear interpolation only when other ESA CCI SM products are incorporated as model predictors. The best results across all metrics, for all tested combinations, are obtained in the case where combined and active ESA CCI SM RF downscaled products with incorporating ESA CCI SM products in the first RF model are used. The combination that uses combined and passive products and the one using just the combined product follow closely. Additionally, shifting NDVI values to obtain better matching with in situ data also introduces improvements across all ESA CCI SM combinations. The average improvement after NDVI shift in RMSE, $R^2$ and MAE is 0.0004 m$^3$/m$^3$, 0.0038 and 0.0001 m$^3$/m$^3$, respectively.

When comparing the downscaling using RF model with other two ESA CCI SM products and the downscaling using bilinear interpolation, the first one outperforms the other for most combinations. The differences in validation metrics vary with the combination of used ESA CCI SM predictors. Differences can be marginally small or as large as 0.0036 m$^3$/m$^3$, 0.0399 and 0.0027 m$^3$/m$^3$ for RMSE, $R^2$ and MAE, respectively. The metrics' differences are also not largely affected by the NDVI shift. These results suggest that the main work is done by the second RF model, while the method of downscaling of ESA CCI SM products in the first step has limited effect. All downscaling methods introduce additional errors which are not always successfully modeled in the second RF model. This is particularly seen in the case of the downscaling step without other ESA CCI SM predictors, where it actually deteriorates the quality of the final output.

In all tested combinations, the use of several ESA CCI SM products yielded better results than the use of only one product. Although the combined product is generated from active and passive data, it turns out that some variability between these three is left unaccounted for in the assimilation process. In congregation with other predictors, such variability can be successfully exploited by the two-step downscaling procedure. Since the RF downscaling procedure is complex and with significant requirements for memory and processing power, in some cases, the bilinear interpolation method might be preferred over it in the data engineering step. This can especially happen over larger areas, where RF model might become too heavy for standard uses and when simplicity of the bilinear interpolation is useful.

Considering that the usage of active and combined ESA CCI SM downscaled product (with ESA CCI SM products in RF model 1) has proved to be the best solution for generating high-resolution soil moisture maps, all additional validation was done only for that one, instead for all tested predictor combinations.

## 6.2. Predictor Importance

The predictor's relative importance was determined using the percentage of increase in RMSE that its omittance produces. The predictor being tested is omitted from all processing steps (from both RF models) and the RMSE is afterwards determined using the same validation technique as before. All ESA CCI SM products were grouped and treated as a single predictor to generalize interconnections and dependencies that exist between the two RF models. The NDVI shift was also considered and the relative importance was determined in both cases—with and without using NDVI shift.

Figure 8 shows that the group of ESA CCI SM products is by far the most important predictor. If omitted, at least two times bigger RMSE increase is to be expected, compared to the omittance of other predictors. Day of year (DOY) and NDVI can be classified as medium important predictors with increase of RMSE between 4% and 5% and the remaining ones are the least important predictors, having

the increase of less than 1%. The NDVI shift slightly increased NDVI, DOY and $LST_{DAY}$ importance, but it also decreased the relative importance of all the other predictors.

Such predictor importance corresponds to the observed correlation coefficient between the in situ soil moisture observations and the used predictors, except for the $LST_{DAY}$ predictor. $LST_{DAY}$ shows significant negative correlation with the in situ data ($-0.55$), but its importance is the smallest among all the predictors. The explanation for this can be that this is the result of the existence of two LST predictors. This way, the importance of each LST predictor is independently small, yet they have their share in the performance of the prediction model. On the other hand, the correlation coefficient of the DOY predictor is minor ($-0.20$), but it is the one of the top-three predictors by importance. Because some of the used predictors have delayed effect on the soil moisture content, such behavior can be better modeled by introducing the time information explicitly through DOY predictor. DOY information also helps the RF model to capture the yearly weather seasons, which have strong effect on the soil moisture content.

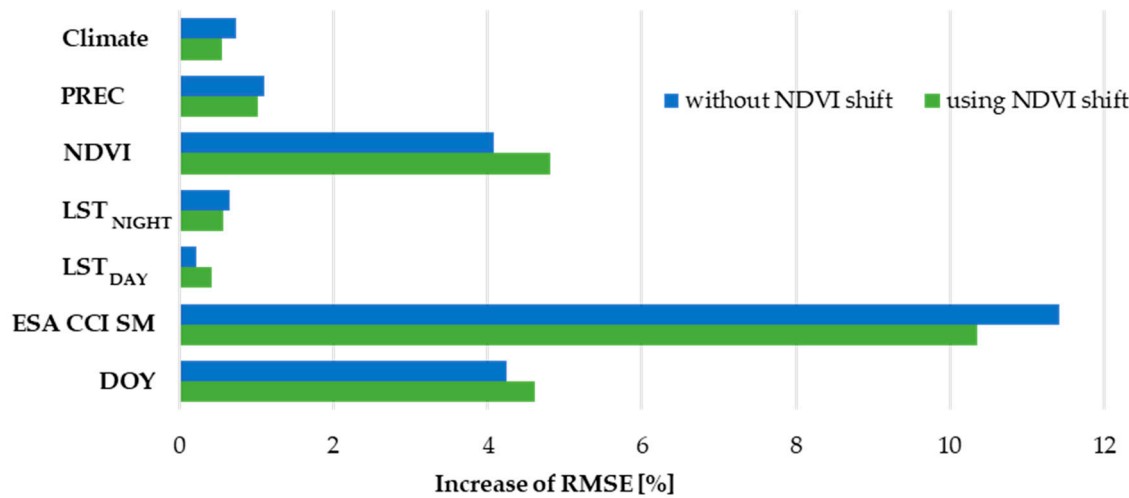

**Figure 8.** Relative variable importance based on increase of RMSE.

### 6.3. Spatial Patterns of the High-Resolution Soil Moisture Maps

The validation metrics have been calculated for every station independently (all metrics are available in Appendix A, Table A1). The calculated metrics have a wide range of values, with RMSE ranging 0.0182–0.1102 $m^3/m^3$, $R^2$ ranging 0.0000–0.9674 and MAE ranging 0.0141–0.0825 $m^3/m^3$ for results without the NDVI shift. When the NDVI shift is included, the metrics' upper boundaries are slightly improved, with RMSE of 0.0186–0.1065 $m^3/m^3$, $R^2$ of 0.0000–0.9694 and MAE of 0.0139–0.0795 $m^3/m^3$. Individually, if the RMSE threshold is defined as 0.04 $m^3/m^3$, only 19 stations (with and without the NDVI shift) reach this threshold. This is rather low performance, although it needs to be noted that low RMSE is in a way compensated by high $R^2$. From the stations that fail to reach the threshold, two thirds of them have $R^2$ higher than 0.7 and almost half of them have $R^2$ values higher than 0.8. No statistical relationship between RMSE and $R^2$ values has been detected. The number of observations per stations does not affect the metrics either, which suggests that the reason for such behavior needs to be examined regarding its spatial and climate characteristics.

Spatial patterns are examined by creating the bubble plots of the previously calculated metrics per station. As shown in Figure 9, the larger values across all metrics are more present in the coastal regions, while the values are generally smaller in the mainland. The NDVI shift has limited effect on spatial patterns, where individual values are changed, but the trend along the coast is still present. Such spatial patterns are attributed to the California's relief (Figure 1), which in a way creates a "wall" that limits the influence of the ocean and its effect on the precipitation and other climate conditions.

The climate zones by Köppen–Geiger are also differentiated by the "mountain wall". The ocean heavily impacts areas that are located between the coast and the mountain wall, while the mainland behind the mountains has its own climate conditions. The proximity of the ocean affects the soil moisture patterns because it influences precipitation amounts and the precipitation is taken as the main source of the soil moisture change. Near the coast, the tropical climate is present with larger precipitation amounts. Since the perception is used as a predictor in the model, the changes of SSM are successfully modeled near the coast (high $R^2$ values). Nevertheless, the RMSE has larger values because the precipitation has a spatial resolution of only 5 km and its additional improvements are necessary in order to reduce the RMSE values. On the contrary, the arid climate in the mainland with lower precipitation amounts has small variations of SSM content, which are not primarily caused by precipitation. Such variations are not modeled properly (lower $R^2$ values), but the remaining predictors still model total amount of SSM reasonably well, which provides lower RMSE values.

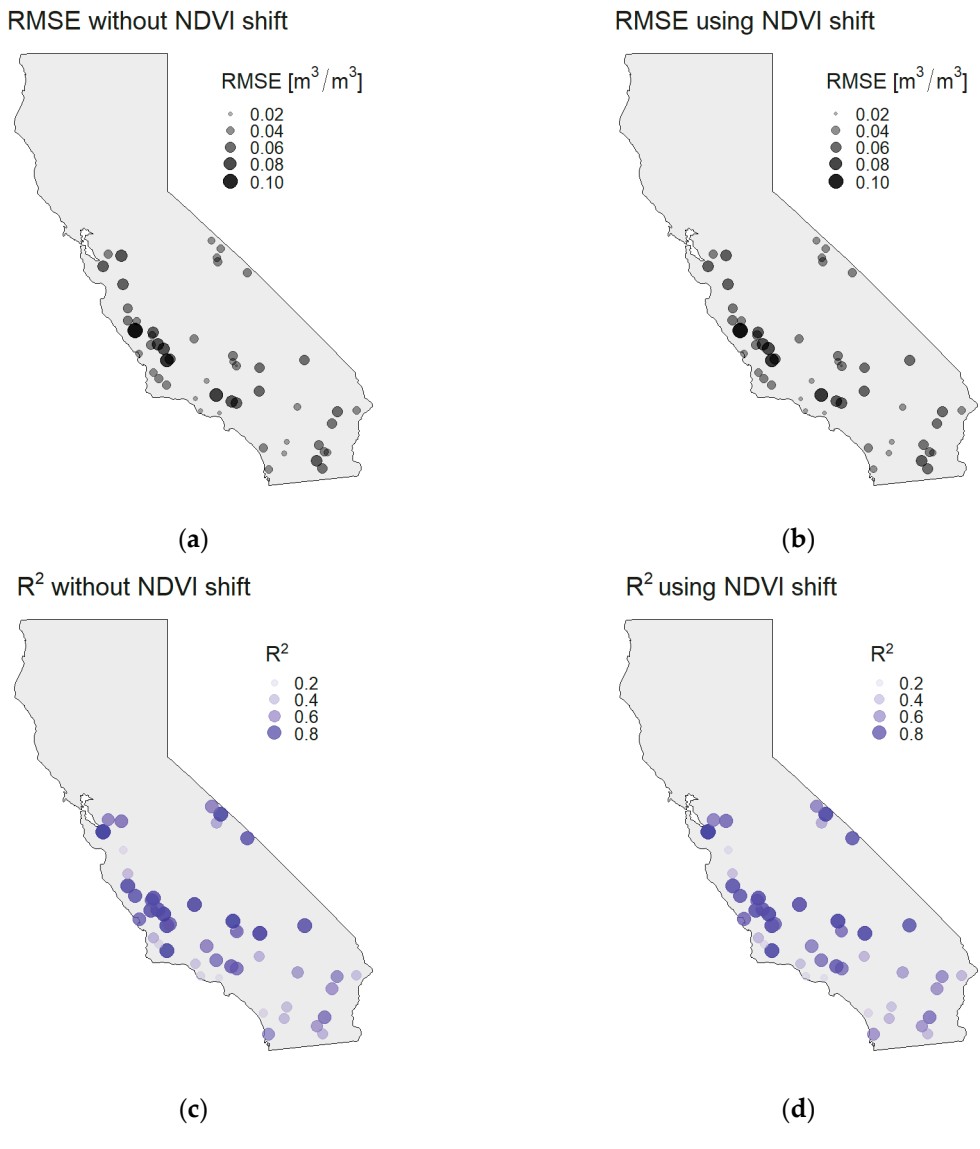

**Figure 9.** *Cont.*

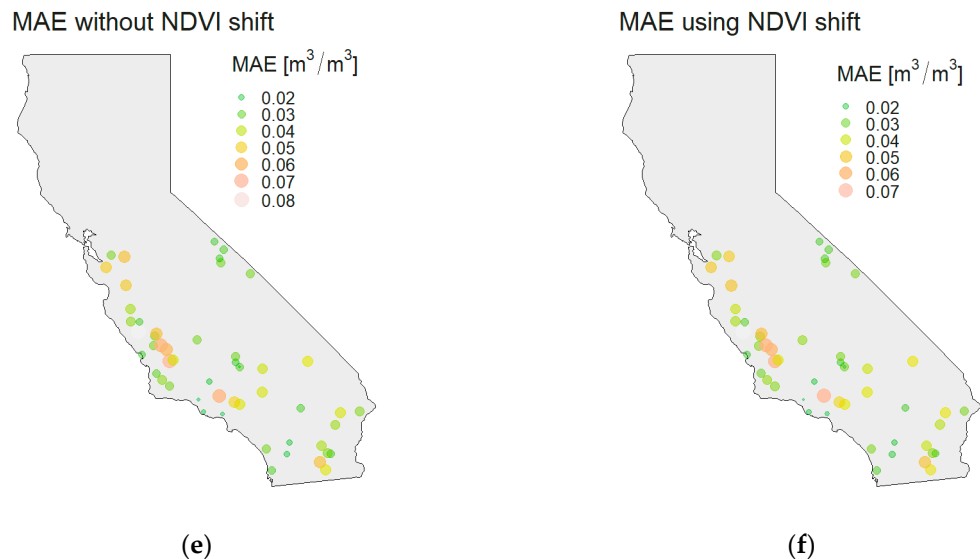

(**e**)                                                   (**f**)

**Figure 9.** Spatial distribution of the calculated metrics presented by bubble plots without and with using NDVI shift: (**a**,**b**) RMSE; (**c**,**d**) $R^2$; and (**e**,**f**) MAE.

These assumptions are confirmed by creating the boxplot charts of calculated metrics per climate type. Since spatial patterns are almost identical for results with and without NDVI shift, the boxplots of metrics are determined only for the results without NDVI shift. It can be clearly seen in Figure 10 that Climate Class B (desert and semi-arid climates) that covers most of the mainland has lower values of all metrics when compared to Climate Class C (tropical/megathermal climates). It can be said that the model is more accurate in the desert and semi-arid climate. This is more due to the lesser variations in the soil moisture than the efficiency of the model. For the tropical/megathermal climates, the situation is reversed, that is, the model is more efficient, but the overall accuracy of the downscaled data is lower. Although the climate is included in the predictor set as an attempt to differentiate such areas, it is clear that this approach was not successful. The potential solution might be to use this information as a way to split up the area of interest into segments and to build independent models for each segment separately. A similar approach has been done by soil types, and it proved to be useful [32]. In addition, the additional predictors should be checked and included in order to reduce RMSE in tropical/megathermal climates, but with the preservation of high $R^2$ value.

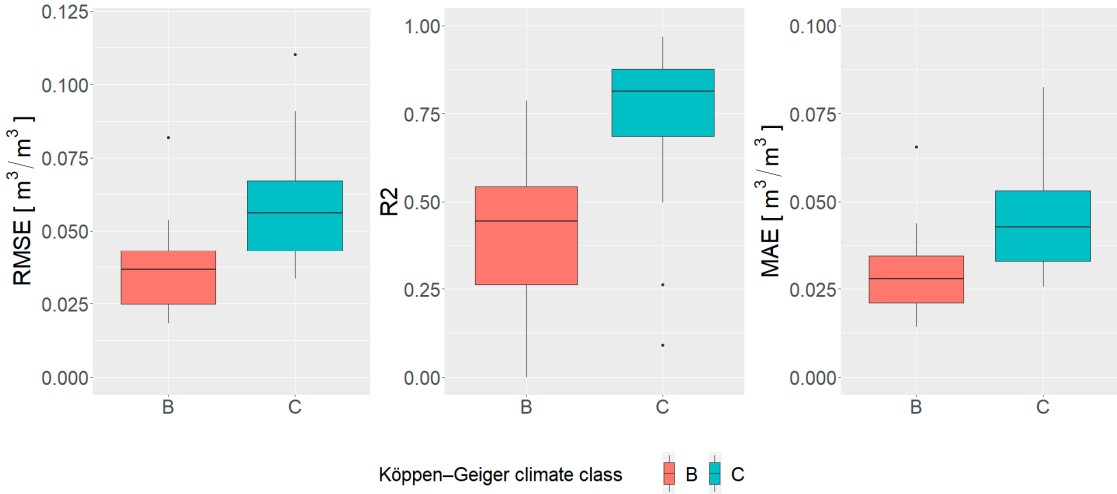

**Figure 10.** Boxplot of the calculated metrics (without NDVI shift) per stations differentiated by climate classes.

The spatial patterns are further examined regarding the land cover type by creating the boxplot charts per land cover type. For that purpose, the MODIS Land Cover Type Yearly L3 Global 500 m 2016 and its University of Maryland (UMD) classification scheme were used. Unfortunately, only a few land cover classes are well represented. Grassland, open shrubland and barren or sparsely vegetated land cover classes are represented with 25, 11 and 10 stations, respectively. All other land cover classes have three or fewer stations across the study area and therefore have to be left out from the boxplot, since there are insufficient data for analysis. As shown in Figure 11, the performance of the remaining three land cover classes varies per class. As expected, the barren or sparsely vegetated class and open shrubland class have lower RMSE and MAE than stations over grasslands. On the other hand, $R^2$ values over grasslands are very high (above 0.75) while $R^2$ of the other two land cover classes are below 0.5. Such values correspond to the findings of the other researches, where modeling soil moisture content becomes harder as the amount of vegetation increases [1].

Unfortunately, this highlights the main disadvantage of using PBO_H2O soil moisture network for modeling soil moisture content. Since it is a GPS based method for determining soil moisture content, only the land cover classes that provide open-sky conditions necessary for GPS signal can be used. Consequently, not all land cover classes (especially forests and other dense vegetation) can be covered. Therefore, since such land cover classes are not used during the training of the RF model, it is unlikely that the model will be able to provide good performance for these areas.

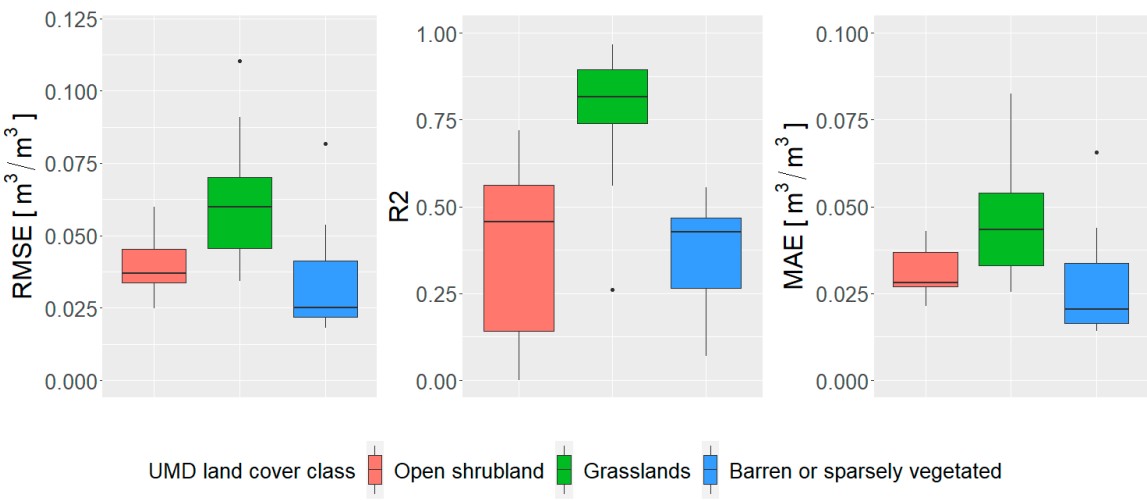

**Figure 11.** Boxplot of the calculated metrics (without NDVI shift) per stations differentiated by MODIS land cover classes (UMD classification scheme).

*6.4. Temporal Patterns of the High-Resolution Soil Moisture Maps*

The temporal line plot for each station is available in Appendix B Figure A1. Only the extreme ones are discussed in this section. The extreme values of RMSE and $R^2$ for the stations with more than 280 observations in 2016 are presented in Figure 12.

The modeled soil moisture is generally smoother when compared to the in situ values. Small variations are usually omitted, but most of the larger leaps are still successfully captured by the model. For most of the stations with larger RMSE values, the modeled soil moisture content is smaller than the in situ soil moisture content. The differences are larger than 0.2 m³/m³ and they mostly occur after the change in the soil moisture content (due to the precipitation). This suggests that, even though the precipitation is included in the predictors set, the model does not exploit that information well enough. The main advantage of the modeled soil moisture is that there are no missing data, which is a significant problem for some stations where several months of missing data occur. Although such results cannot be confirmed, by looking at the plotlines, the change of soil moisture contents in the time windows with missing in situ data seems to be reasonable, without unusual spikes and downs.

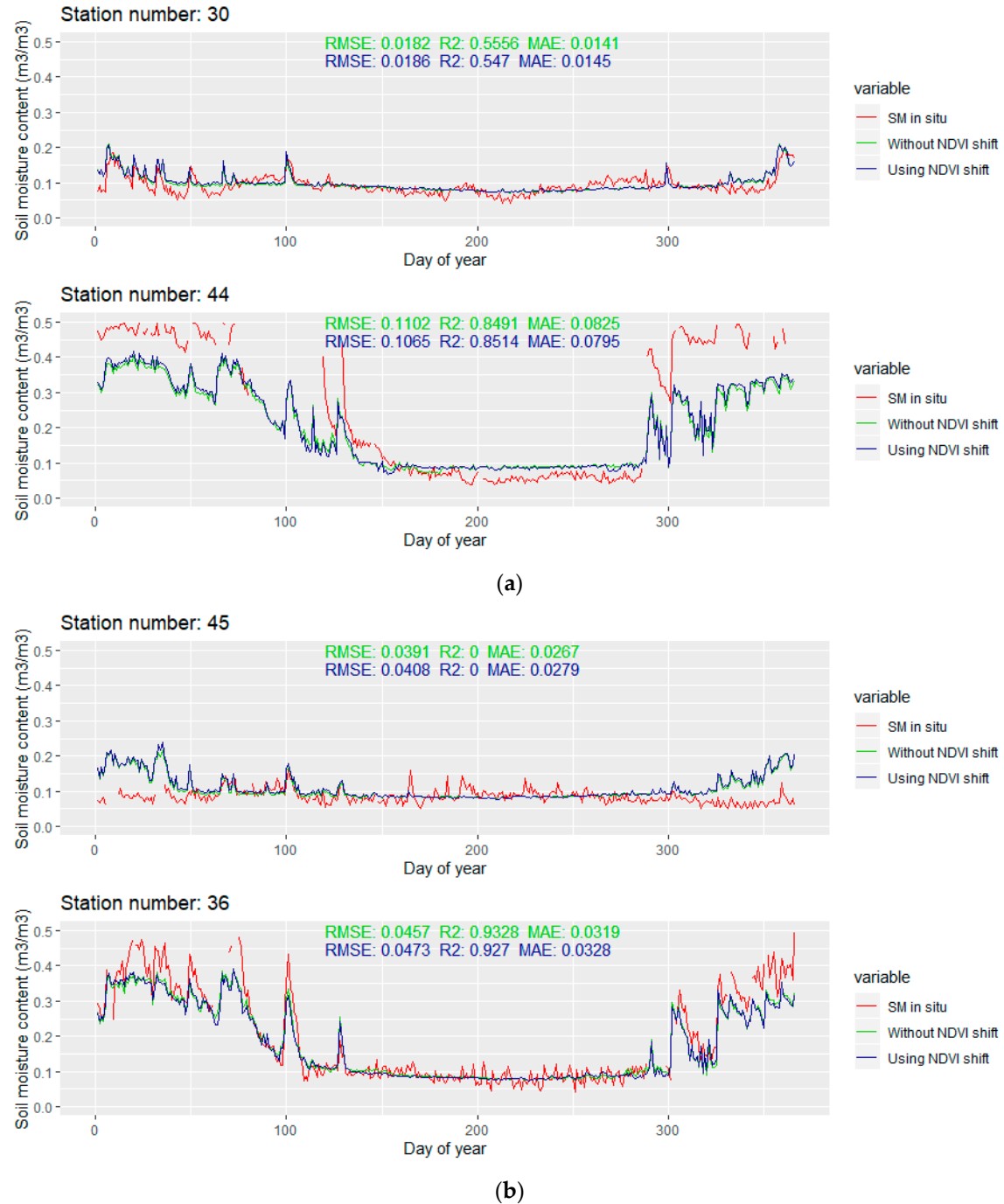

**Figure 12.** The temporal line plots of the stations with extreme values of RMSE and $R^2$: (**a**) the minimum value of RMSE for Station 30 and the maximum value of RMSE for Station 44; and (**b**) the minimum value of $R^2$ for Station 45 and the maximum value of $R^2$ for Station 36.

Temporal patterns of the soil moisture were also evaluated for every month of 2016. The in situ and the predicted soil moisture content have range of variations, which differ by month, as shown in Figure 13. The in situ variations are smallest in the June–September period, while they are larger throughout the rest of the year. Such patterns can also be observed in the predicted soil moisture content, although those variations are smaller. This corresponds to the smoothing introduced by the prediction model. Such behavior is especially present in the June–September period.

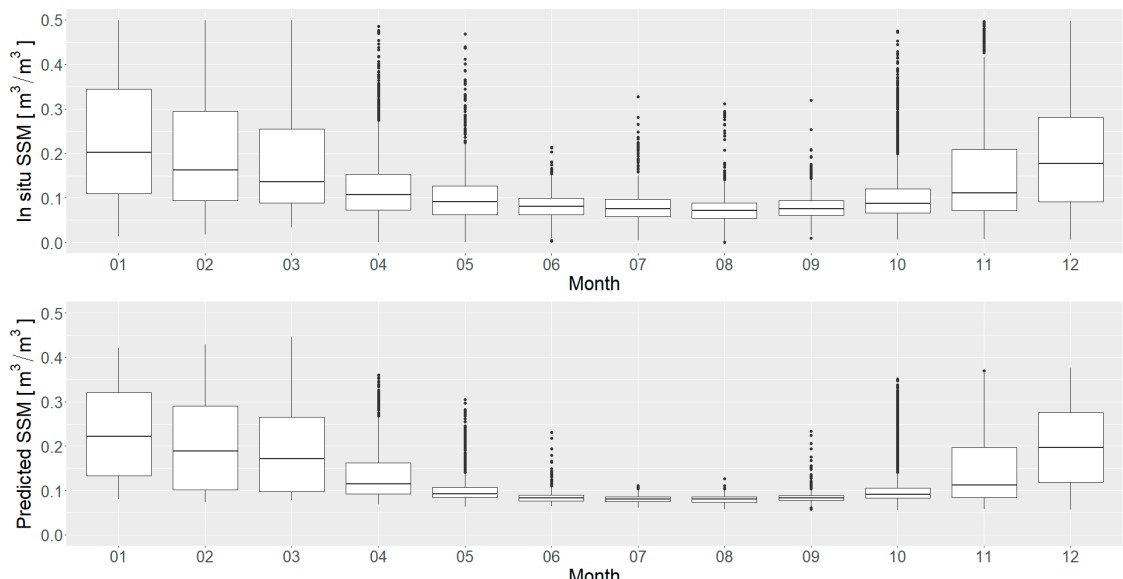

**Figure 13.** The boxplot charts of the in situ soil moisture observations and the five-fold cross-validations results (without NDVI shift), for every month of 2016.

The temporal monthly patterns were further inspected by calculating monthly validation metrics (Figure 14). As expected, the model efficiency differs during the year, with June–September RMSE and MAE being significantly lower than for the rest of the months. $R^2$ is very low for the same months and with improvements during the rest of the year.

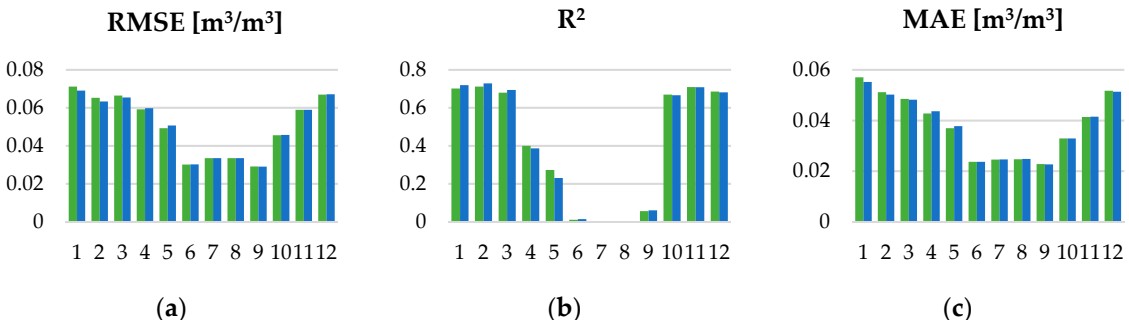

**Figure 14.** Validation metrics calculated over each month of 2016: (**a**) RMSE; (**b**) $R^2$; and (**c**) MAE;.

The monthly metrics correspond to the observed monthly variations. The months with smaller variations in SSM content in June–September period have relatively large measurement uncertainty. This is due to the reported accuracy of the in situ observations of 0.04 m$^3$/m$^3$. Although no observation weights are included in the RF model, this uncertainty is successfully characterized by RF model, producing smooth observations for this period. The predicted smooth lines mostly correspond to the average of in situ observations for these days. RMSE values under 0.04 m$^3$/m$^3$ for these months indicate that the smooth averaged predictions of the soil moisture are sufficiently accurate. On the other hand, the months with larger variations in SSM content show systematic behavior. The prediction model can capture this behavior successfully, which is proved by the strong correlation. Unfortunately, only the bigger changes are successfully modeled, while the smaller ones are omitted due to the same effect that exists in the June–September period. This produces larger RMSE values for these periods. Additionally, more snow and cloud cover is expected in these months, thus large areas of missing data might occur. The gap filling step is less effective in these conditions, which could also have an impact on higher values of RMSE for these months.

All these differences in RMSE and $R^2$ over the year imply that the model has limitations regarding the soil moisture variations that can be successfully captured. These limitations are primarily caused by the accuracy of the soil moisture observations, which needs to be accounted for in the prediction model.

*6.5. Independent Validation of the High-Resolution Soil Moisture Maps*

Independent validation of the generated soil moisture maps was performed using the in situ soil moisture observations available from the SCAN and USCRN soil networks. These datasets were pulled from the ISMN data archive. Only the top-soil soil moisture observations at 5 cm depth were taken into the account, which makes 15 SCAN stations and 6 USCRN stations available over the study area (Figure 15). Each station has daily soil moisture measurements for the whole 2016, resulting in 7686 soil moisture observations being available for the validation. For each station, the same validation metrics were calculated as in the five-fold cross-validation.

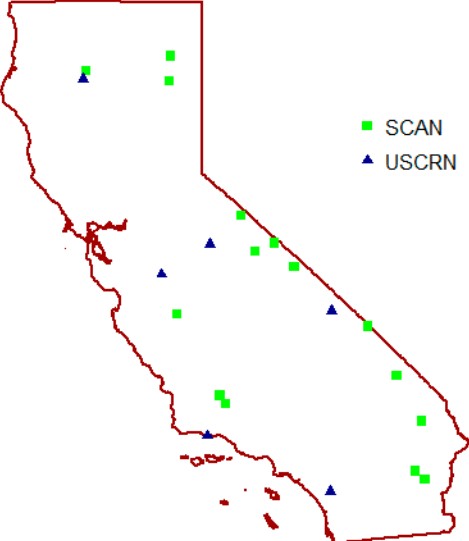

**Figure 15.** Spatial distribution of the SCAN and USCRN soil moisture stations over the study area used for the independent validation.

The calculated metrics (Table 3) show rather poor results of the produced high-resolution soil moisture maps against SCAN and USCRN stations data. Very large RMSE and MAE and low $R^2$ values over both networks suggest that these two sources are not at all comparable with the high-resolution soil moisture maps. The reason for this can be attributed to the differences in the soil moisture measurement depth and the mismatch of data sources regarding their spatial resolution. Both SCAN and USCRN in situ soil moisture observations provide soil moisture content at certain soil depth (5 cm depth used for the validation). Since second RF model is modeled using PBO_H2O in situ data that represent top-soil interval of 0–5 cm, the output high-resolution soil moisture maps also correspond to the 0–5 cm soil depth interval. Additionally, SCAN and USCRN in situ measurements are point measurements, while the output soil moisture maps have spatial resolution of 1 km. Point-scale in situ measurements need to be upscaled to the desired spatial resolution [46,47] before these two datasets can be compared. Such discrepancy in the spatial resolution is not present while building the second RF model since PBO_H2O spatial resolution of 1 km matches the spatial resolution of the used predictors. Conversion from the point-scale in situ measurement at certain depth to the interval measurements of 1 km spatial resolution is beyond the scope of this research. That is why the authors believe that the poor validation metrics against SCAN and USCRN stations should not be taken as a true quality assessment of the created high-resolution soil moisture maps and that the five-fold cross-validation metrics provide more realistic quality assessment.

**Table 3.** Validation metrics of the produced high-resolution soil moisture obtained by using SCAN and USCRN soil moisture stations data.

| Netwok | Stations Count | Without NDVI Shift | | | Using NDVI Shift | | |
|---|---|---|---|---|---|---|---|
| | | RMSE $[m^3/m^3]$ | $R^2$ | MAE $[m^3/m^3]$ | RMSE $[m^3/m^3]$ | $R^2$ | MAE $[m^3/m^3]$ |
| SCAN | 15 | 0.0885 | 0.2584 | 0.0689 | 0.0891 | 0.2631 | 0.0694 |
| USCRN | 6 | 0.0908 | 0.5666 | 0.0757 | 0.0938 | 0.5633 | 0.0776 |

## 7. Conclusions

The methodology used in this study proved to be a good solution for creating high resolution surface soil moisture maps over the area of California, USA. Within the research, downscaled SSM maps were produced for 2016. The output product spatial resolution was improved to 1 km. The proposed approach also considered filling data gaps as an initial step of the procedure, which in the end produced continuous product in both temporal and spatial domains. The filling of the missing data was performed using the universal kriging in the spatial domain and by applying spline fitting and interpolation in the temporal domain. The daily datasets without missing data were then used to produce 1 km soil moisture maps using two-step procedure. The first step was used as a data-engineering tool and its output was used as the input for the second step. The ESA CCI SM products and PBO_H2O in situ soil moisture observations were used as a main data input in the congregation with NDVI, LST, precipitation and climate zones as auxiliary datasets. The validation metrics were calculated for several tested models to determine the optimal one.

Comparison of the model results and soil moisture observations from SCAN and USCRN soil networks yielded rather poor results, which suggest that these two sources are not comparable due to the differences in the soil moisture measurement depth and the spatial resolution. That is why the model performance was evaluated through five-fold cross-validation. The best prediction model obtained soil moisture with RMSE of 0.0518 $m^3/m^3$ and $R^2$ of 0.7312, which is comparable to the results made by similar studies [19].

Our study found that both bilinear interpolation and RF downscaling procedure could be used as a data engineering tool for providing the additional predictors in soil moisture prediction models. As the calculated validation metrics indicate, the optimal soil moisture prediction model uses RF downscaled combined and active ESA CCI SM products in congregation with other auxiliary datasets and in situ soil moisture observations. The models that use bilinear interpolation as a data engineering tool provided results that are only marginally deteriorated. This is because the RF regression model in the second step does most of the work. The study showed that, when downscaling one type of ESA CCI SM product, the remaining two types of ESA CCI SM products in congregation with other predictors should be used. Although a combined product is generated from active and passive data, it turns out that, in the assimilation process, some variability between these three is left unaccounted for. The study also implemented the NDVI shift, due to its delayed effect on the SSM, in order to boost its correlation with in situ soil moisture observations. This proved to be useful, with the improvements of all metrics across all model variations.

The study also highlights the pros and cons of using PBO_H2O in situ soil moisture observations for soil moisture downscaling. Since it is a GPS based method, only the land cover classes that provide open-sky conditions necessary for GPS signal are available. Consequently, not all land cover classes (especially forests and other dense vegetation) can be covered. The accuracy of the soil moisture observations limits the amount of variations that could be successfully modeled in the downscaling procedure. On the other hand, the observations' spatial resolution of 1 km matches the commonly desired SSM output. This way, there is no need for in situ upscaling procedure which can induce additional prediction errors.

By analyzing the spatial patterns of the validation metrics, it is concluded that the model performances vary for different climate zones, even though the climate is included as a model predictor. The climate information should be further inspected and possibly used as a way to divide the area of interest into segments. The spatial patterns have also been examined regarding the land cover class. The model performed best over the barren or sparsely vegetated and open shrubland areas and it had lower performance over grasslands. Individually per station, higher RMSE value is followed by high $R^2$ value and vice versa. The model also has temporal variability, with lower RMSE values and low $R^2$ values over the June–September period, and higher RMSE and $R^2$ values for other months.

Unfortunately, PBO_H2O in situ soil moisture observations are no longer available because the project ended in 2017. Similar projects in the future are encouraged and welcomed, especially with improved accuracy of the SSM observations. In the meantime, the more common point-scale in situ observations might be usable, but this requires further testing. If so, the methodology can easily be transferable to other study sites, as long as some of the in situ soil moisture observations are available. All other used datasets are globally available, except the precipitation dataset which should be replaced by some of the alternative dataset.

Some additional sources of the used products (e.g., The Copernicus Global Land Service for NDVI and LST) should also be considered and incorporated within the gap filling procedure. The gap filling can be additionally improved by using the kriging with an external drift instead of the universal kriging. Beside these improvements, new predictors, such as soil characteristics, albedo, topography, etc., can be added to the model in the future, which could further improve soil moisture predictions. Additionally, downscaling of the precipitation dataset could also be considered, since its effect on the SSM variability is significant.

**Author Contributions:** All authors worked equally on the research conceptualization, investigation, discussion, and data modeling. J.K. was involved in data preparation and case study implementation. Ž.C., D.M., N.S. and N.B. participated in the writing of the manuscript, however J.K. took the lead. All authors have read and agreed to the published version of the manuscript.

**Funding:** This research received no external funding.

**Acknowledgments:** This study was supported by the Serbian Ministry of Education, Science and Technological Development, project TR 36020.

**Conflicts of Interest:** The authors declare no conflict of interest.

**Appendix A**

**Table A1.** Five-fold cross-validation metrics aggregated over 10 independent splits for each station.

| Station ID | KG Climate Class | Land Cover Class (UMD) | Obs. Count | Without NDVI Shift | | | Using NDVI Shift | | |
|---|---|---|---|---|---|---|---|---|---|
| | | | | RMSE [m$^3$/m$^3$] | $R^2$ | MAE [m$^3$/m$^3$] | RMSE [m$^3$/m$^3$] | $R^2$ | MAE [m$^3$/m$^3$] |
| 1 | B | 10 | 239 | 0.0527 | 0.6981 | 0.0390 | 0.0538 | 0.7097 | 0.0399 |
| 2 | B | 7 | 311 | 0.0365 | 0.6583 | 0.0281 | 0.0341 | 0.6468 | 0.0269 |
| 3 | B | 16 | 281 | 0.0538 | 0.4969 | 0.0437 | 0.0546 | 0.4744 | 0.0442 |
| 4 | C | 10 | 325 | 0.0674 | 0.6202 | 0.0539 | 0.0648 | 0.6385 | 0.0516 |
| 5 | B | 16 | 323 | 0.0253 | 0.4741 | 0.0202 | 0.0264 | 0.4875 | 0.0216 |
| 6 | B | 7 | 366 | 0.0341 | 0.1623 | 0.0275 | 0.0308 | 0.2232 | 0.0252 |
| 7 | B | 9 | 364 | 0.0445 | 0.7491 | 0.0341 | 0.0442 | 0.7477 | 0.0340 |
| 8 | B | 13 | 363 | 0.0419 | 0.3098 | 0.0327 | 0.0420 | 0.3086 | 0.0327 |
| 9 | C | 7 | 365 | 0.0490 | 0.0911 | 0.0386 | 0.0508 | 0.0453 | 0.0402 |
| 10 | B | 16 | 366 | 0.0247 | 0.4520 | 0.0209 | 0.0251 | 0.4276 | 0.0211 |

**Table A1.** *Cont.*

| Station ID | KG Climate Class | Land Cover Class (UMD) | Obs. Count | Without NDVI Shift | | | Using NDVI Shift | | |
|---|---|---|---|---|---|---|---|---|---|
| | | | | RMSE [m³/m³] | R² | MAE [m³/m³] | RMSE [m³/m³] | R² | MAE [m³/m³] |
| 11 | C | 10 | 364 | 0.0516 | 0.6515 | 0.0373 | 0.0532 | 0.6577 | 0.0386 |
| 12 | B | 16 | 365 | 0.0208 | 0.2470 | 0.0159 | 0.0216 | 0.2399 | 0.0163 |
| 13 | C | 9 | 366 | 0.0568 | 0.6773 | 0.0430 | 0.0563 | 0.6787 | 0.0436 |
| 14 | B | 16 | 330 | 0.0222 | 0.3164 | 0.0177 | 0.0231 | 0.2736 | 0.0187 |
| 15 | B | 16 | 366 | 0.0375 | 0.4366 | 0.0344 | 0.0356 | 0.4890 | 0.0325 |
| 16 | C | 7 | 309 | 0.0334 | 0.6184 | 0.0273 | 0.0314 | 0.5999 | 0.0256 |
| 17 | C | 9 | 305 | 0.0646 | 0.5597 | 0.0507 | 0.0652 | 0.5554 | 0.0511 |
| 18 | C | 10 | 366 | 0.0623 | 0.7811 | 0.0434 | 0.0629 | 0.7564 | 0.0429 |
| 19 | C | 10 | 351 | 0.0701 | 0.8165 | 0.0490 | 0.0660 | 0.8465 | 0.0465 |
| 20 | B | 16 | 365 | 0.0219 | 0.4168 | 0.0148 | 0.0212 | 0.4439 | 0.0139 |
| 21 | C | 10 | 362 | 0.0837 | 0.7437 | 0.0648 | 0.0848 | 0.7546 | 0.0664 |
| 22 | C | 7 | 317 | 0.0598 | 0.4977 | 0.0430 | 0.0595 | 0.5056 | 0.0426 |
| 23 | C | 10 | 277 | 0.0837 | 0.8414 | 0.0600 | 0.0859 | 0.8255 | 0.0615 |
| 24 | C | 10 | 215 | 0.0413 | 0.9163 | 0.0330 | 0.0417 | 0.9163 | 0.0333 |
| 25 | B | 16 | 226 | 0.0818 | 0.0706 | 0.0657 | 0.0809 | 0.0594 | 0.0643 |
| 26 | B | 7 | 366 | 0.0247 | 0.7200 | 0.0214 | 0.0243 | 0.7197 | 0.0211 |
| 27 | B | 7 | 366 | 0.0433 | 0.3089 | 0.0355 | 0.0437 | 0.2903 | 0.0358 |
| 28 | B | 7 | 363 | 0.0369 | 0.5036 | 0.0296 | 0.0380 | 0.4769 | 0.0305 |
| 29 | C | 10 | 358 | 0.0553 | 0.9235 | 0.0416 | 0.0540 | 0.9114 | 0.0405 |
| 30 | B | 16 | 366 | 0.0182 | 0.5556 | 0.0141 | 0.0186 | 0.5470 | 0.0145 |
| 31 | C | 10 | 364 | 0.0428 | 0.7395 | 0.0299 | 0.0456 | 0.7133 | 0.0314 |
| 32 | B | 7 | 366 | 0.0319 | 0.1234 | 0.0248 | 0.0317 | 0.1050 | 0.0249 |
| 33 | C | 10 | 260 | 0.0909 | 0.8956 | 0.0695 | 0.0850 | 0.9015 | 0.0649 |
| 34 | C | 10 | 361 | 0.0544 | 0.8733 | 0.0445 | 0.0547 | 0.8596 | 0.0438 |
| 35 | C | 10 | 363 | 0.0600 | 0.8106 | 0.0426 | 0.0588 | 0.8049 | 0.0419 |
| 36 | C | **10** | 340 | 0.0457 | 0.9328 | 0.0319 | 0.0473 | 0.9270 | 0.0328 |
| 37 | B | 10 | 340 | 0.0344 | 0.7878 | 0.0255 | 0.0335 | 0.7994 | 0.0250 |
| 38 | C | 10 | 319 | 0.0739 | 0.9224 | 0.0590 | 0.0760 | 0.9126 | 0.0597 |
| 39 | C | 10 | 359 | 0.0457 | 0.8704 | 0.0334 | 0.0454 | 0.8801 | 0.0333 |
| 40 | C | 10 | 290 | 0.0732 | 0.8691 | 0.0634 | 0.0720 | 0.8619 | 0.0621 |
| 41 | C | 12 | 361 | 0.0403 | 0.9081 | 0.0327 | 0.0431 | 0.8891 | 0.0353 |
| 42 | C | 12 | 284 | 0.0399 | 0.8071 | 0.0312 | 0.0462 | 0.7651 | 0.0352 |
| 43 | C | 6 | 366 | 0.0660 | 0.8704 | 0.0544 | 0.0646 | 0.8797 | 0.0535 |
| 44 | C | 10 | 285 | 0.1102 | 0.8491 | 0.0825 | 0.1065 | 0.8514 | 0.0795 |
| 45 | B | **7** | 352 | 0.0391 | 0.0000 | 0.0267 | 0.0408 | 0.0000 | 0.0279 |
| 46 | C | 13 | 239 | 0.0485 | 0.8772 | 0.0360 | 0.0502 | 0.8786 | 0.0371 |
| 47 | B | 7 | 366 | 0.0473 | 0.4567 | 0.0381 | 0.0486 | 0.4478 | 0.0398 |
| 48 | C | 10 | 348 | 0.0627 | 0.2619 | 0.0514 | 0.0621 | 0.2670 | 0.0508 |
| 49 | C | 12 | 364 | 0.0433 | 0.8493 | 0.0327 | 0.0425 | 0.8563 | 0.0324 |
| 50 | C | 10 | 152 | 0.0649 | 0.9674 | 0.0506 | 0.0625 | 0.9694 | 0.0496 |
| 51 | B | 16 | 239 | 0.0426 | 0.0886 | 0.0321 | 0.0430 | 0.0779 | 0.0321 |
| 52 | C | 10 | 344 | 0.0363 | 0.5600 | 0.0272 | 0.0375 | 0.5487 | 0.0280 |
| 53 | C | 10 | 343 | 0.0699 | 0.7566 | 0.0538 | 0.0630 | 0.8017 | 0.0489 |
| 54 | C | 8 | 178 | 0.0419 | 0.7131 | 0.0331 | 0.0430 | 0.7241 | 0.0338 |
| 55 | C | 10 | 353 | 0.0373 | 0.9127 | 0.0277 | 0.0366 | 0.9242 | 0.0270 |
| 56 | C | 10 | 365 | 0.0345 | 0.7206 | 0.0255 | 0.0349 | 0.7032 | 0.0260 |

## Appendix B

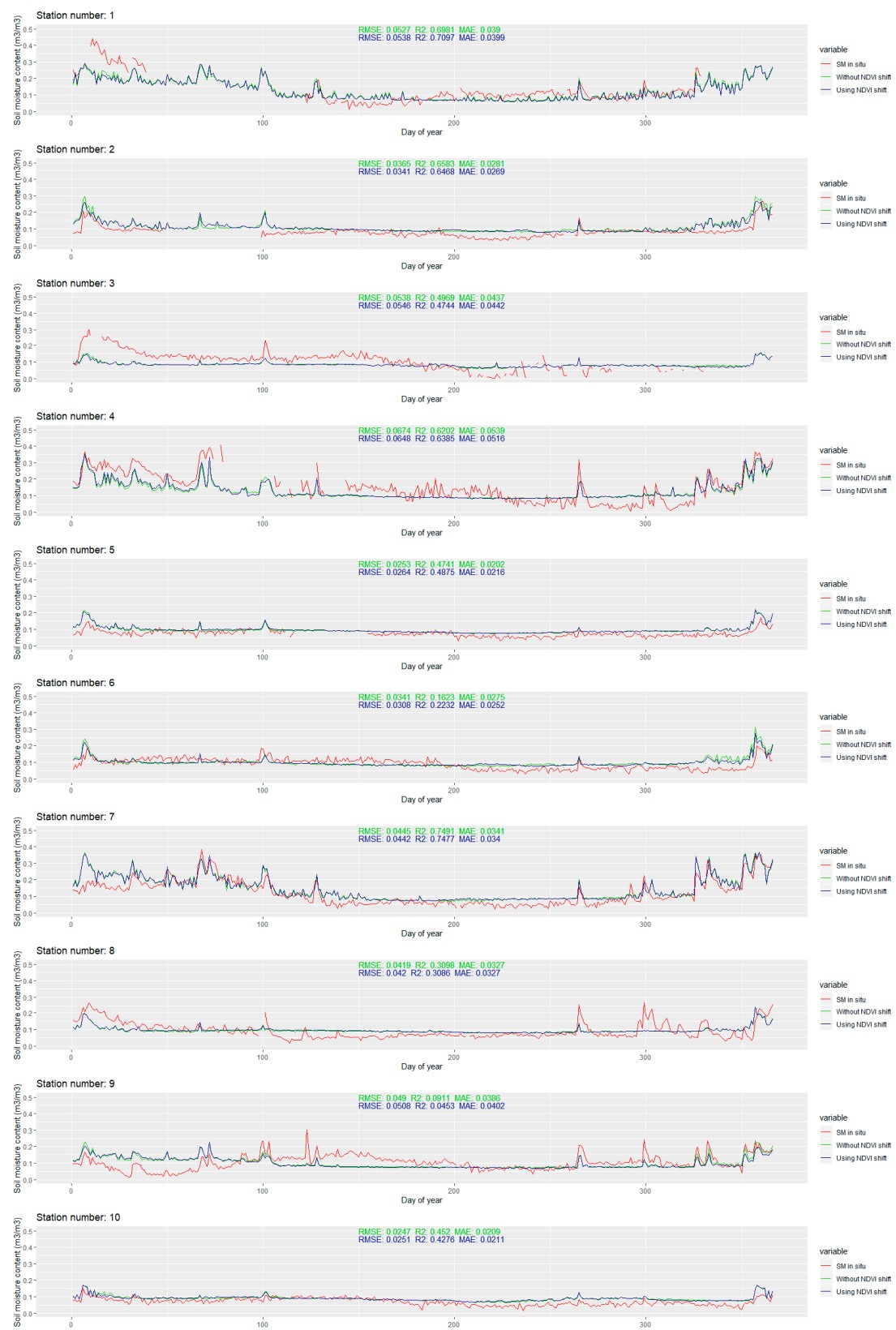

**Figure A1.** *Cont.*

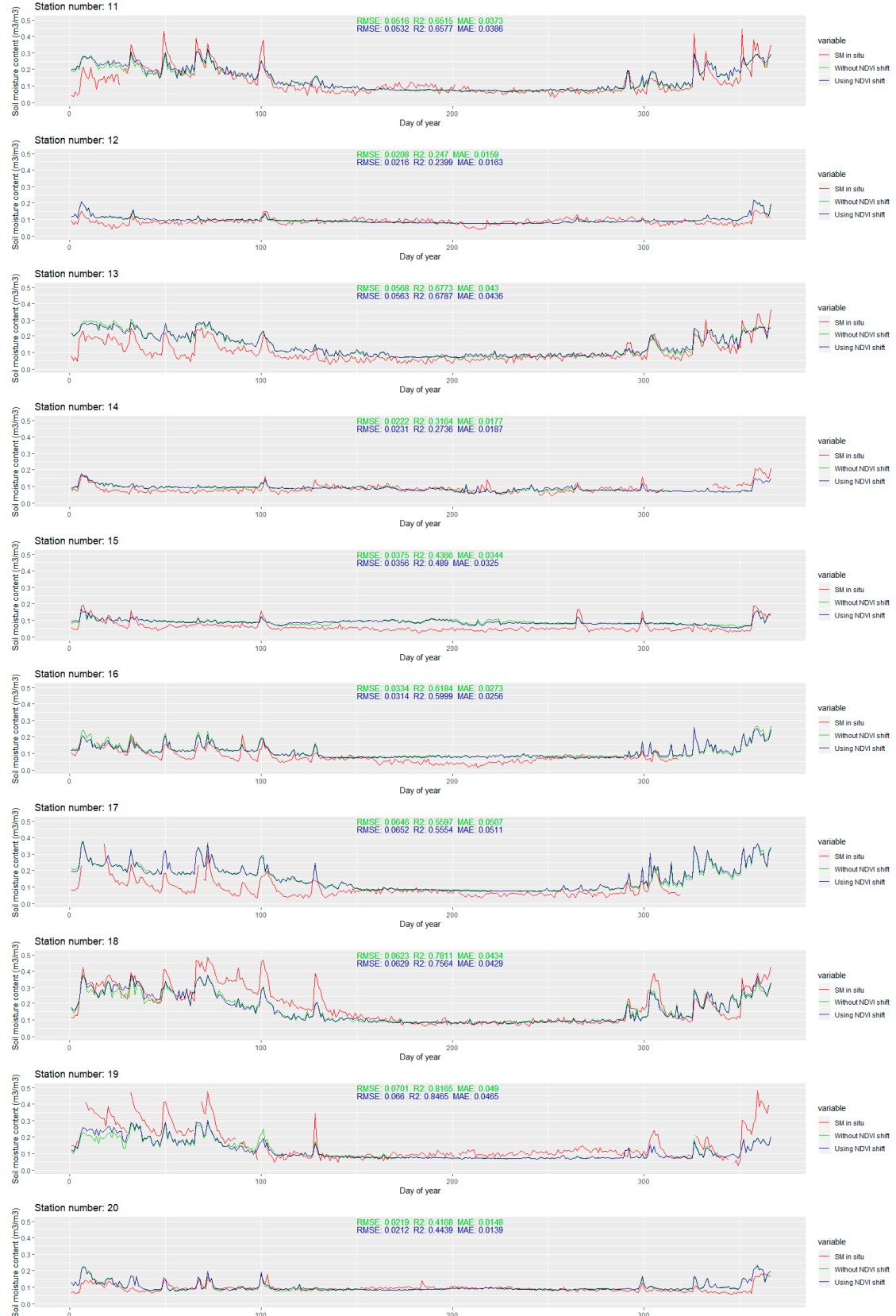

**Figure A1.** *Cont.*

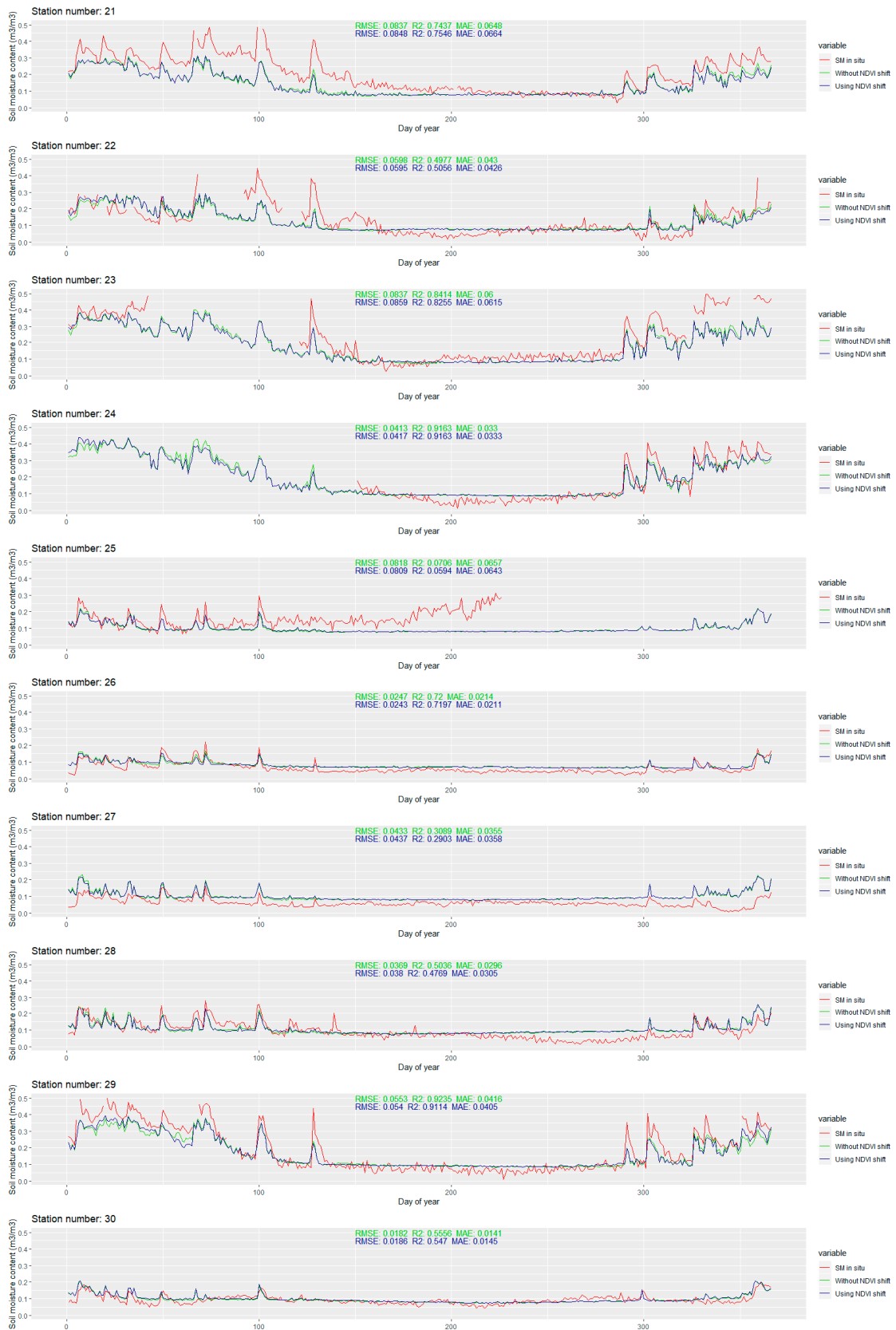

**Figure A1.** *Cont.*

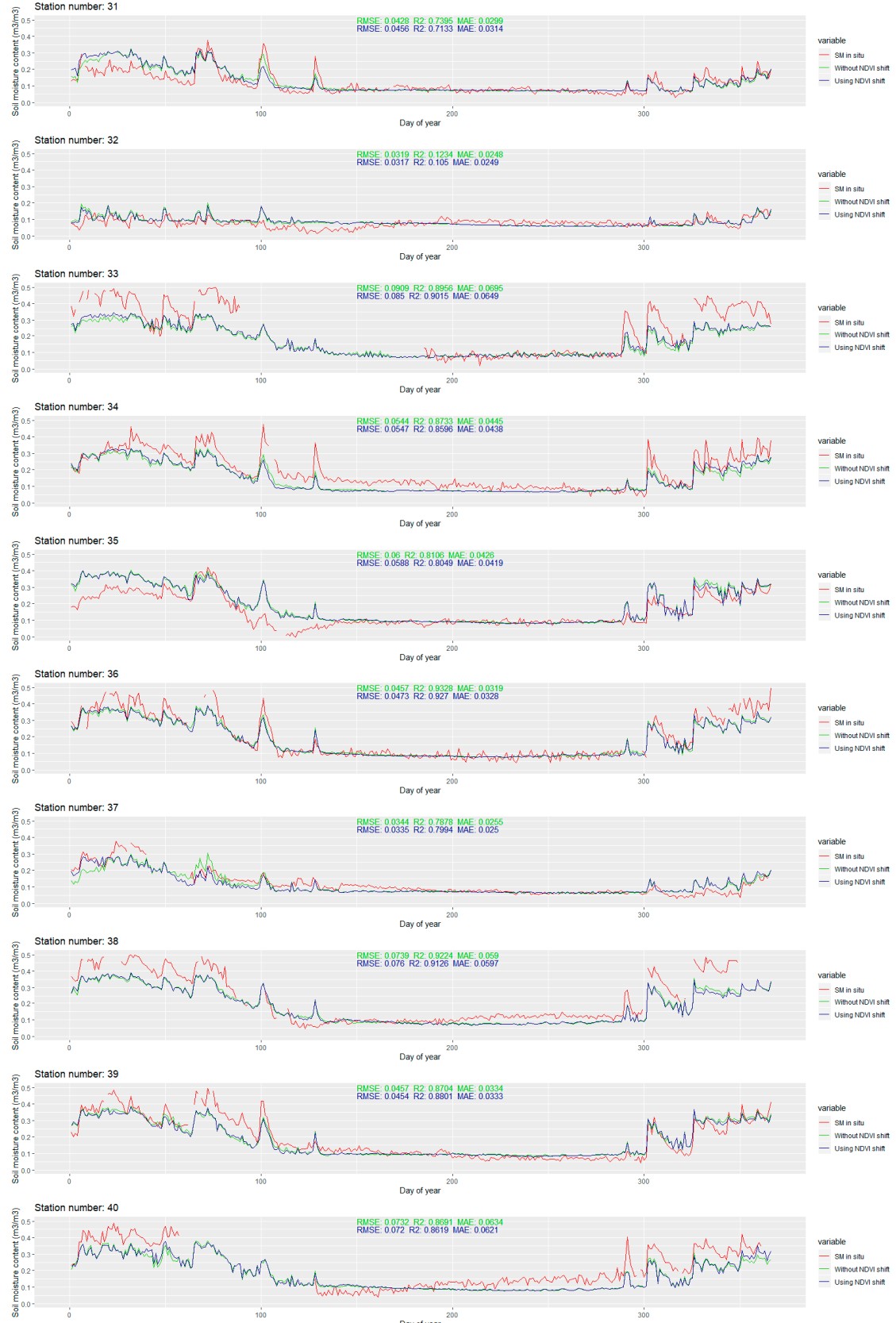

**Figure A1.** *Cont*.

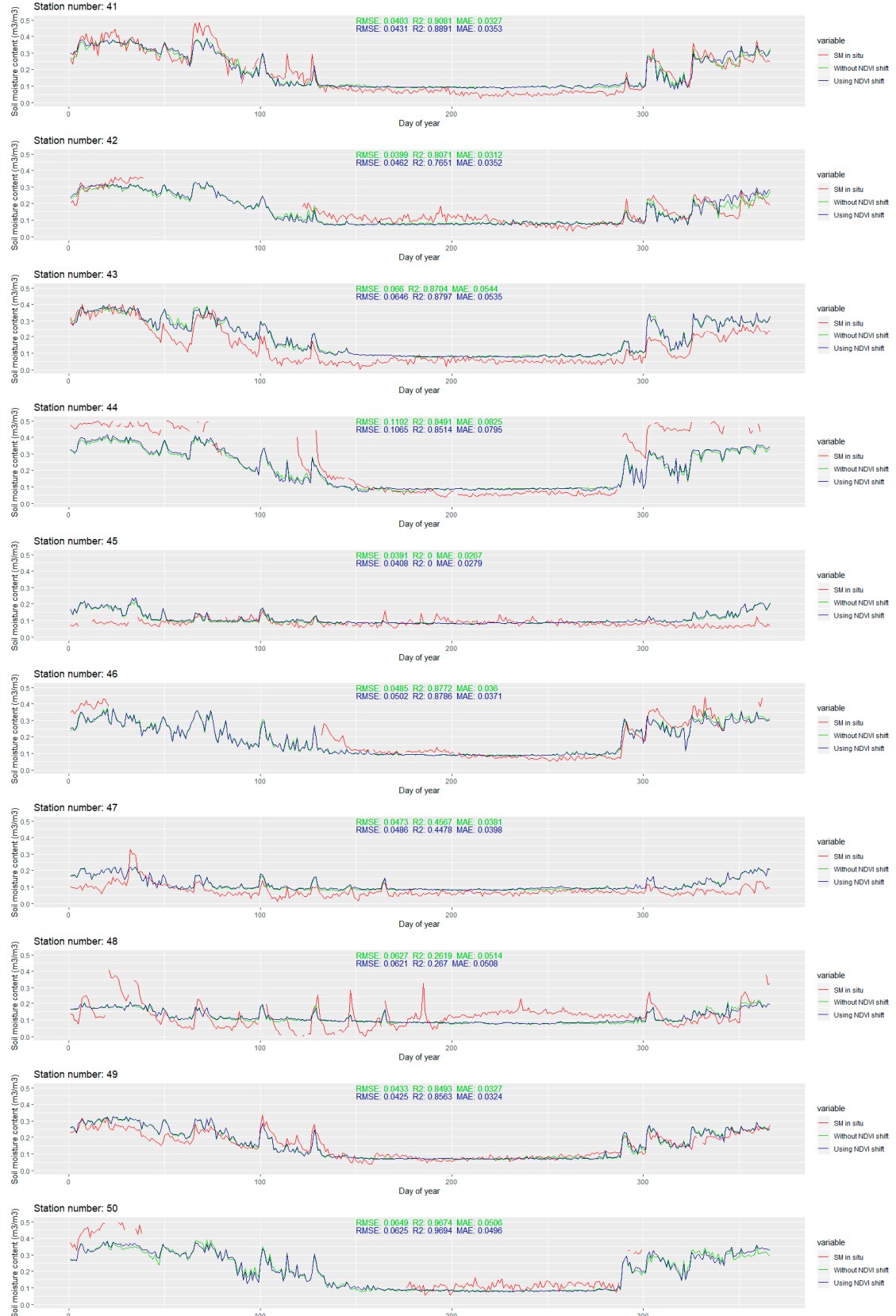

**Figure A1.** *Cont*.

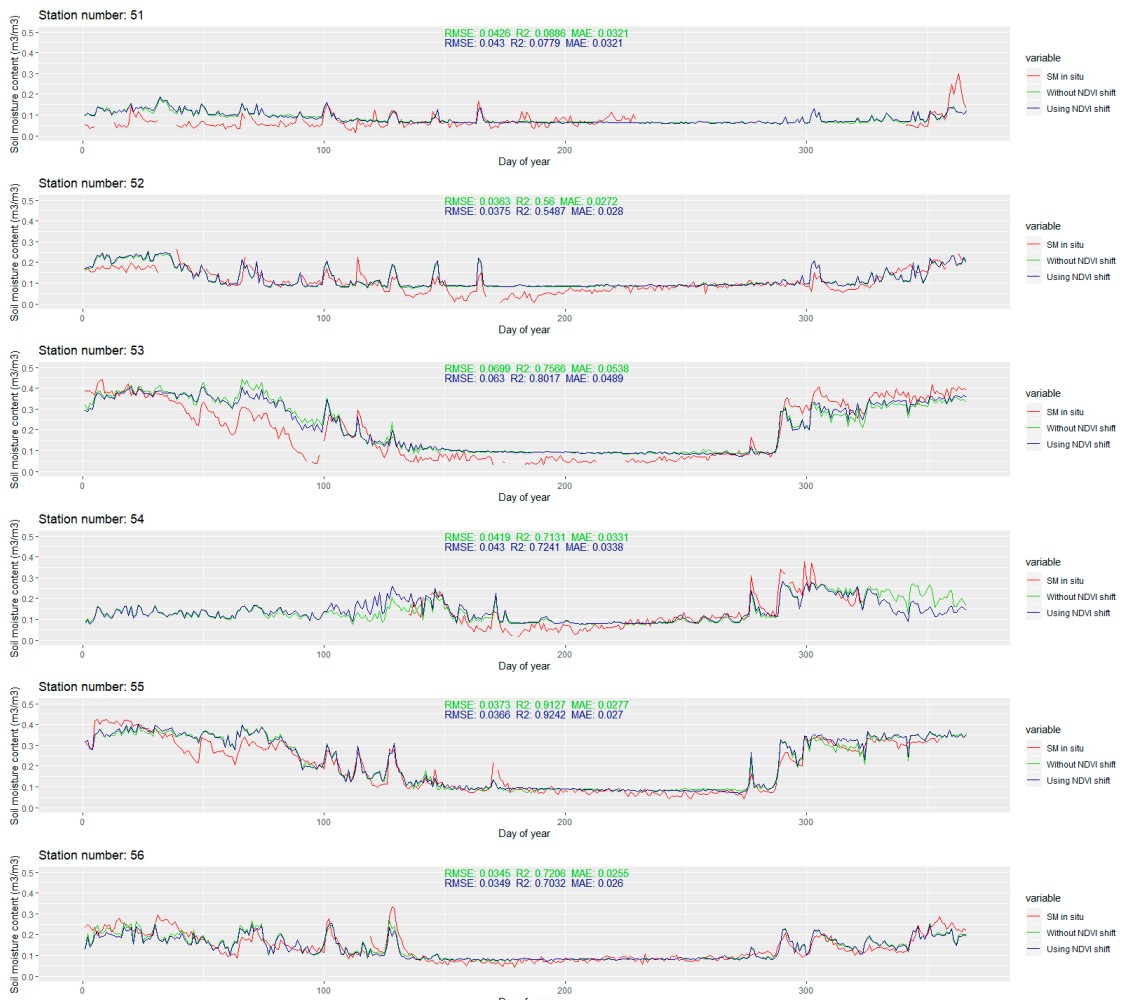

**Figure A1.** Temporal dynamics of modeled soil moisture against in situ values for each station in the study area.

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
