# Peer review of "New Downscaling Approach Using ESA CCI SM Products for Obtaining High Resolution Surface Soil Moisture"

_remotesensing, doi:10.3390/rs12071119_

Round 1

Reviewer 1 Report

I attach a PDF file including my comments.

Reviewer 2 Report

This manuscript presents a Random Forest machine learning-based soil moisture downscaling work. A two-step random forest scheme is applied to the ESA CCI soil moisture products, including active, passive, and combined datasets. Multiple auxiliary data are used and a 1 km soil moisture data over California is generated. Overall, this manuscript is relatively well structured, and the results are shown in a good manner. However, the reviewer has several concerns and suggestions.

  1. To obtain the spatially and temporally continuous downscaled 1 km soil moisture data, the data gaps are filled for the various datasets, especially the MODIS NDVI. However, the authors also note that there are large errors in the filled NDVI data that can further impact the downscaled accuracy. It is not very clear why such a filling method is chosen. In figure 4, it is clear that high NDVI values are smoothed out that can lead to a smooth soil moisture time series.
  2. A 2-step random forest method is used for downscaling soil moisture. However, comparing the three groups in Table 3, the first-step random forest only leads to marginally improved (or even deteriorated) accuracy compared to the bilinear disaggregation method. More discussion will be needed for addressing this issue.
  3. Since a 1-km soil moisture data is available for California. It is curious to see an independent validation (besides the 5-fold) of the downscaled soil moisture over independent sites, such as the SCAN and USCRN from the International Soil Moisture Network.

A few minor comments follow below.

  1. Why a DOY (time information) provide such high importance? More discussion is needed.
  2. Are Table 2 and Table 1 evaluating the same downscaled data against different reference data? In Table 1, the meanings of ‘without’ and ‘with’ are not clearly indicated.
  3. Figure 4, no legend.
  4. Please increase the font size in Figures 6, 8, 9, and 11. The color bar can also be enlarged.
  5. Line 416: under-measurements?
  6. Line 432: ‘bellow’ to ‘below’

Round 2

Reviewer 1 Report

I would like to thank the authors for addressing almost all my comments throughout this paper. I am glad that the paper has been extensively modified.

I believe the paper could now be accepted for publication in its current form.

Author Response

We are very thankful to the reviewer for participating in the reviewing process, especially in these troubling Corona times. We are looking forward to some new future collaborations!

Authors

Reviewer 2 Report

The authors have largely improved the presentation of their work. And it now can be accepted. Some minor grammar errors need to be checked. For example, line 181-182 (used climate in the downscaling soil moisture), line 721 ('Point in situ measurements' to 'Point-scale in-situ measurements').

Author Response

We are very thankful to the reviewer for participating in the reviewing process, especially in these troubling Corona times. Beside the grammar errors highlighted by the reviewer, we have checked the rest of the document for the grammar and spelling errors as well and made additional corrections.